# A Comparative Study on the Lysosomal Cation Channel TMEM175 Using Automated Whole-Cell Patch-Clamp, Lysosomal Patch-Clamp, and Solid Supported Membrane-Based Electrophysiology: Functional Characterization and High-Throughput Screening Assay Development

**DOI:** 10.3390/ijms241612788

**Published:** 2023-08-14

**Authors:** Andre Bazzone, Maria Barthmes, Cecilia George, Nina Brinkwirth, Rocco Zerlotti, Valentin Prinz, Kim Cole, Søren Friis, Alexander Dickson, Simon Rice, Jongwon Lim, May Fern Toh, Milad Mohammadi, Davide Pau, David J. Stone, John J. Renger, Niels Fertig

**Affiliations:** 1Nanion Technologies, Ganghoferstr. 70a, 80339 Munich, Germanyvalentin.prinz@nanion.de (V.P.); soren.friis@nanion.de (S.F.); 2RIGeL-Regensburg International Graduate School of Life Sciences, University of Regensburg, 93053 Regensburg, Germany; 3SB Drug Discovery, West of Scotland Science Park, Glasgow G20 0XA, UK; alexander.dickson@sbdrugdiscovery.com (A.D.); simon.rice@sbdrugdiscovery.com (S.R.);; 4Cerevel Therapeutics, 222 Jacobs St, Cambridge, MA 02141, USA; jongwon.lim@cerevel.com (J.L.); mayfern.toh@cerevel.com (M.F.T.); david.stone@cerevel.com (D.J.S.); john.renger@cerevel.com (J.J.R.); 5Assay Works, Am BioPark 11, 93053 Regensburg, Germany

**Keywords:** TMEM175, ion channels, Parkinson’s disease (PD), screening technologies, drug development, lysosomes, solid supported membrane-based electrophysiology (SSME), SURFE^2^R, automated patch-clamp (APC), lysosomal patch-clamp

## Abstract

The lysosomal cation channel TMEM175 is a Parkinson’s disease-related protein and a promising drug target. Unlike whole-cell automated patch-clamp (APC), lysosomal patch-clamp (LPC) facilitates physiological conditions, but is not yet suitable for high-throughput screening (HTS) applications. Here, we apply solid supported membrane-based electrophysiology (SSME), which enables both direct access to lysosomes and high-throughput electrophysiological recordings. In SSME, ion translocation mediated by TMEM175 is stimulated using a concentration gradient at a resting potential of 0 mV. The concentration-dependent K^+^ response exhibited an I/c curve with two distinct slopes, indicating the existence of two conducting states. We measured H^+^ fluxes with a permeability ratio of P_H_/P_K_ = 48,500, which matches literature findings from patch-clamp studies, validating the SSME approach. Additionally, TMEM175 displayed a high pH dependence. Decreasing cytosolic pH inhibited both K^+^ and H^+^ conductivity of TMEM175. Conversely, lysosomal pH and pH gradients did not have major effects on TMEM175. Finally, we developed HTS assays for drug screening and evaluated tool compounds (4-AP, Zn as inhibitors; DCPIB, arachidonic acid, SC-79 as enhancers) using SSME and APC. Additionally, we recorded EC_50_ data for eight blinded TMEM175 enhancers and compared the results across all three assay technologies, including LPC, discussing their advantages and disadvantages.

## 1. Introduction

TMEM175 is a lysosomal cation channel involved in regulating lysosomal pH, lysosomal membrane potential, and autophagy. It was originally known as a potassium leak channel [1,2,3,4], but recent advances suggest that it also facilitates proton transport. For example, it was shown that the overexpression of TMEM175 in lysosomes led to increased proton conductance, thus indicating that TMEM175 may be a proton channel as well as a potassium channel [1]. In this study, the knock-out cell line exhibited impaired lysosomal pH stability, a finding subsequently validated in another report [5]. A most recent publication suggests that under physiological conditions, TMEM175 acts as a proton-gated, proton-selective channel rather than a constitutively active potassium-selective channel [6], with the purpose of counteracting the over-acidification of lysosomes, where the channel’s H^+^ conductance is described as essential for normal lysosome function. A recent review has provided a comprehensive overview of TMEM175, covering its functional properties, structural data, physiological role, regulation, and pathology [7]. While further research is needed to fully understand the role of TMEM175 in lysosomal proton transport and acidification, the results presented in our study provide evidence that TMEM175 may function as a proton channel in addition to its well-established role as a potassium channel.

TMEM175 has gained increasing attention over the past years [8] because of its putative involvement in neurodegenerative diseases such as Parkinson’s [9,10] and Lewy Body Dementia [11]. The link between impaired lysosomal function and Parkinson’s disease (PD) is not new [12]. However, TMEM175 emerged as a relevant potential therapeutic target for the treatment of PD [13,14]. Therefore, understanding the pharmacological relevance of TMEM175 and the development of screening assays for compounds affecting TMEM175 activity could lead to the development of novel therapeutics, potentially also for other neurodegenerative diseases.

In the initial phases of drug discovery, the primary goal is to identify compounds that elicit a response against a validated target. Compound libraries with 100,000+ compounds are not uncommon. Therefore, in the primary screening phases, ultra-high throughput in terms of number of compounds per day are required. If the molecular structure and the active site of the protein is known, in silico methods, such as molecular docking, could be an option [15,16]. The alternative is laboratory assays, compatible with high-throughput screening (HTS) standards. Examples of HTS-compatible methods for TMEM175 are binding assays [17], Flipr-based Thallium flux assays [18,19], radioactive [3] or non-radioactive rubidium ion flux assays [20], or Flipr-based screening utilizing voltage-sensitive membrane dyes [21]. All methods except for the binding assays yield a functional read-out in response to TMEM175 stimulation, and give a crude estimate of which compounds could be worthwhile pursuing, in so-called heat maps. Therefore, secondary screening follows for compound validation and characterization.

Automated patch-clamp (APC) is a commonly used method for secondary screening and hit validation on ion channels. The technique combines real-time, high-quality data with throughput capabilities of 10,000+ data points per day. The original, manual patch-clamp approach is considered the gold standard for obtaining information on ion channels and their effectors [22], but is extremely labor intensive with low data throughput. Because of its tremendous potential for ion channel drug screening, attempts to automate the procedure took off in the early 2000s [23,24]. Several automated patch-clamp platforms became commercially available and some of them were successfully used for recordings on TMEM175, for instance, the SyncroPatch (Nanion Technologies, München, Germany) [25] and the Qube (Sophion Biosciences, Ballerup, Denmark) [26]. A recent study highlights the applicability and versatility of APC platforms [27].

Since TMEM175 is a lysosomal channel, and APC platforms are optimized for whole-cell recordings, cell lines [2,3] and methods [28] have been developed to overexpress TMEM175 in the plasma membrane. In this configuration, the luminal, i.e., intra-lysosomal side of TMEM175 faces the extracellular side of the plasma membrane. A relevant question here is whether the unnatural lipid and protein environment of the plasma membrane or the different protein orientation could have effects on the apparent compound pharmacology or efficacy. Recently, a direct interaction between lysosomal LAMP proteins and TMEM175 was reported [29], potentially altering the protein/drug interaction sites of TMEM175 in its native environment compared to the plasma membrane localization.

To enable the measurement of lysosomal ion channels in its natural environment, lysosomal patch-clamp (LPC) has been employed. To make the lysosomes accessible to patch-clamp recordings, they have to be fused into larger structures, for example, using vacouline-1 treatment [30]. Manually patch-clamping chemically enlarged lysosomes is extremely tedious and of low throughput, but is still considered the gold standard for investigating TMEM175 conductance in its native environment. However, an effect on the integrity of the lysosomal membrane and lysosomal physiology is evident due to a vacuolin-1-induced alkalization of lysosomal pH [31], with potential effects on TMEM175 and the compound pharmacology or efficacy.

In this study, we employ solid supported membrane-based electrophysiology (SSME). SSME recordings reveal electrogenic events, such as ionic currents, through the conductive charging of the sample membrane. SSME does not offer the possibility to control the membrane voltage throughout the recordings, meaning that there is no holding potential. The voltage is usually close to 0 mV at the time point of the read-out. Chemical gradients are employed to elicit currents originating from the target protein, in contrast to APC, where the transmembrane currents are measured in response to applying membrane voltage as the driving force. SSME has multiple distinct capabilities compared to patch-clamp: (1) SSME can directly access intracellularly localized target proteins in its natural environments such as TMEM175 in lysosomes and does not require a running cell culture. It can access any target membrane, i.e., proteoliposomes or vesicles from plasma membrane, or organelles such as lysosomes, endoplasmic reticulum, and mitochondria from a wide variety of sources, such as tissue, bacteria, cell lines, and plants. (2) SSME recordings do not rely on live cells. Therefore, ion channel properties can be investigated beyond physiological constraints, e.g., investigating TMEM175 proton and potassium conductance separately. (3) Membrane transporters are often not resolved using whole-cell patch-clamp techniques [32,33,34] due to their low translocation rates, which are typically several orders of magnitude lower compared to ion channels. The large surface of SSME sensors and the cumulative measurement of millions of vesicles enhance the signal-to-noise and facilitate the measurement of membrane transporters. (4) In contrast to LPC, SSME recordings are HTS-compatible. The SSME platforms used in this study, SURFE²R N1 and SURFE^2^R 96SE (Nanion Technologies), offer a data throughput ranging from 150 data points per day to a mid-range of secondary HTS (5–10,000 data points per day), respectively, and are being used for transporter drug screening purposes [35,36].

Current methods for the functional characterization of lysosomal ion channels and transporters have been reviewed recently [37], also including LPC and SSME. Taken together, there is a broad palette of methods for investigating TMEM175 and screening for compounds. They all have their benefits and drawbacks, which poses the following question and challenge: Which method delivers the information that is closest to the actual situation in the lysosomal nano- to micro-environment within the human body? Experimental procedures that potentially could have adverse effects on TMEM175 function, compound pharmacology, and read-out include the overexpression of TMEM175 in cell lines, forcing TMEM175 into the plasma membrane, lysosome enrichment procedures, and lysosome fusion through vacuolin-1.

This study cannot answer these questions, but aims to present the capabilities and results of utilizing SSME for TMEM175 screening. Data obtained with SSME, whole-cell APC, and LPC will be compared with each other and with what is known from the literature.

## 2. Results

The objective of this study is two-fold: to enhance the understanding of the physiological function of TMEM175 and to identify suitable assay conditions for conducting electrophysiological recordings on compounds that accurately represent their potencies in the human body.

To achieve this objective, we performed electrophysiological recordings on lysosomes using SSM-based electrophysiology (SSME) to characterize K^+^ and H^+^ flux and the pH dependence of the channel. Subsequently, we developed a high-throughput screening (HTS)-compatible SSME assay to examine the effects of compounds on TMEM175. We compared the resulting half saturation constants with those obtained from similar assay technologies using the same cell line, namely, the well-established whole-cell APC on TMEM175 located in the plasma membrane, and manual LPC, which is the gold standard in the field. Finally, we tested different assay conditions for their impact on apparent drug potencies, such as K^+^ concentrations, pH gradients, and the orientation of TMEM175 in the target membrane.

All recordings presented in this study were either performed with stable HEK293 cell lines enabling the tetracycline-inducible overexpression of TMEM175 or with lysosomes purified from the same cell lines. We validated the tetracycline-dependent overexpression of TMEM175 using a Western blot (Appendix A).

### 2.1. SSM-Based Electrophysiology Recordings on TMEM175 Localized in Lysosomes

LPC is the gold standard for characterizing lysosomal channels [30]; however, it is very laborious. The whole-cell method has been more commonly utilized to characterize TMEM175 [2,3,28]. One key feature of SSM-based electrophysiology is the possibility for direct measurements on intracellular membranes. Purified lysosomes may be stored at −80 °C for several months before utilization, offering the advantage that SSME experiments can be conducted without the need for a running cell culture.

#### 2.1.1. Purified Lysosomes Contain the Lysosomal Marker LAMP-1

In all of our SSME measurements, we used lysosomal membranes purified according to two established protocols [5,38]. In both cases, cell lysis was performed using a nitrogen Parr bomb. Subsequently, we applied sequential low-speed centrifugation steps [5], or employed ultracentrifugation through a sucrose density gradient [38], which enables the isolation of lysosomal and plasma membrane vesicles (Figure 1A).

To validate that our sample preparation procedure is able to isolate and enrich lysosomes, we performed ELISA assays to determine the concentration of the lysosomal marker LAMP-1 and the plasma membrane marker Na-K-ATPase (Figure 1B). To compare across different samples, we normalized the results to the total protein concentration of the respective sample. We found that the lysosomal samples prepared according to both protocols had very similar concentrations of both LAMP-1 and Na-K-ATPase. In both cases, the LAMP-1 concentration is 3–4 times higher than the concentration of Na-K-ATPase. It must be noted that the ratio between LAMP-1 and Na-K-ATPase concentrations does not represent the ratio of lysosomes to plasma membrane vesicles, since the relative expression or concentration of both proteins within the respective membranes is unknown. However, the enrichment of LAMP-1 indicates that both sample preparation protocols can isolate lysosomes, albeit with some level of impurity from plasma membrane vesicles. For all subsequent SSME recordings, we used samples obtained with the sample preparation protocol described by Schulz et al. [38].

**Figure 1 ijms-24-12788-f001:**
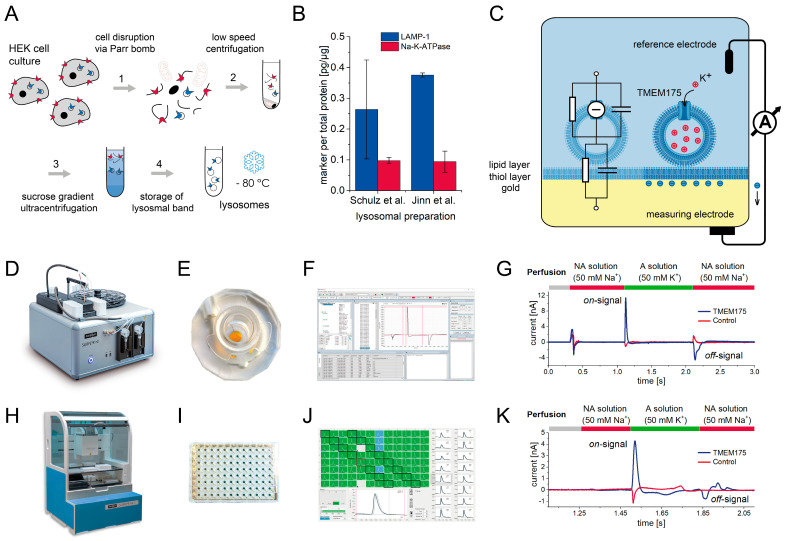
Principles of SSM-based electrophysiology recordings on TMEM175 localized in lysosomes: (**A**) Schematic of the steps performed to purify lysosomes from HEK293 cell culture. The process is based on a protocol by Schulz et al. [38]; (**B**) Marker protein concentrations for lysosomes (LAMP-1) and plasma membrane (Na-K-ATPase) for lysosomal samples purified according to Schulz et al. [38] and Jinn et al. [5] were determined via ELISA. The graph shows the amount of marker proteins normalized to the total protein amount of the respective sample determined via Bradford assay. Average values and standard deviations from N = 2 samples are shown; (**C**) Schematic of lysosomes adsorbed to the SSM on a gold-coated sensor chip. The capacitive read-out is highlighted; (**D**) SURFE^2^R N1 platform for SSME recordings in a single-well format; (**E**) Three-millimeter sensor for the recordings with the SURFE^2^R N1; (**F**) Screenshot of the SURFE^2^R N1 control 1.7.0.2 software; (**G**) Representative current traces recorded with TMEM175 overexpressing lysosomes (blue trace) and control lysosomes (red trace) using the SURFE^2^R N1. The sensors harboring the TMEM175 and the control samples were each loaded with 2.2 µg total protein. SD and average current amplitudes and time constants determined from N = 6 sensors are shown in Table 1. The experiment shows a single solution exchange from a solution containing 50 mM Na^+^ (NA solution, red bars) to a solution containing 50 mM K^+^ (A solution, green bar), which stimulates K^+^ flux through TMEM175 (on-signal). After 1 s, K^+^ is replaced by Na^+^, stimulating K^+^ efflux (off-signal) and restoring initial conditions; (**H**) SURFE^2^R 96SE platform for SSME recordings in HTS format; (**I**) A 96-sensor well-plate for SSME measurements with the SURFE^2^R 96SE. (**J**) Screenshot of the SURFControl96 1.7 software; (**K**) Representative current traces recorded with the SURFE^2^R 96SE. Each sensor was loaded with 0.18 µg total protein. SD and average current amplitudes determined from N = 96 sensors, and average time constants determined from N = 6 sensors are shown in Table 1. Experimental conditions as described in (**G**).

#### 2.1.2. TMEM175 Evokes Currents in SSME Recordings

The technical details and experimental procedures for SSME recordings on TMEM175 are provided in the Appendix A. In brief, the purified sample membrane is attached to the SSM-coated sensor and forms a stable and capacitively coupled membrane system (Figure 1C). In SSME, a solution exchange from a non-activating solution (NA) to an activating solution (A) stimulates substrate translocation. The activating solution provides the substrate. It has been shown that TMEM175 has a higher specificity for K^+^ ions compared to Na^+^ ions [1,28,39]. Based on this knowledge, we used a solution exchange from a Na^+^ containing non-activating solution to a K^+^ containing activating solution to stimulate the K^+^ flux through TMEM175 driven by the K^+^ gradient alone (Standard Assay, Table 2).

For this research, two devices were used (Table 1): the single-well SURFE^2^R N1 platform with high flexibility for assay development (Figure 1D–G), and the SURFE^2^R 96SE platform that enables efficient drug screening using 96-well sensor plates (Figure 1H–K). Technical details for both platforms have been published recently [33] and detailed protocols are available for the SURFE^2^R N1 [34].

Employing the SURFE^2^R N1, using a 1:10 sample dilution before sensor preparation, corresponding to 2.2 µg total protein per sensor, the signal shows an average peak current of (12 ± 3) nA and an average total charge translocation of (219 ± 37) pC (Figure 1G). The rise time τ_1_ of the current is (5.1 ± 1.3) ms. The decay time τ_2_ is (10.7 ± 1.1) ms and significantly faster compared to the typical decay times recorded for transporters that are usually in the range of 30–100 ms, likely due to the high translocation rate of the channel. The current rise and decay are monophasic, indicating the absence of any pre-steady-state currents [40]. However, the fast current decay may be limited by the time resolution of the solution exchange, making it difficult to distinguish ion translocation through the channel and potential pre-steady-state events. In addition, after the current decay, an overshoot of negative amplitude is observed, decaying from negative currents to zero with a system time constant τ_3_ of (285 ± 71) ms. The system time constant is an inherent property of the capacitive membrane system and represents the discharging of the membrane capacitor [32].

Employing the SURFE^2^R 96SE, we optimized for sample consumption and allowed for 12 times less protein use per sensor. The resulting average current amplitude was (4.1 ± 1) nA (Figure 1K), which is a compromise between sample consumption and S/N for efficient measurements in an HTS environment. The time constants of the current rise and decay are in the same range as those determined with the SURFE^2^R N1, at (9.2 ± 1.7) ms and (10.2 ± 0.8) ms, respectively.

#### 2.1.3. Minor Artifact Currents Are Recorded from Lysosomes Purified from Control HEK293 Cells

As a negative control, we performed the same Na^+^/K^+^ exchange experiment on lysosomes purified from HEK293 cells expressing endogenous TMEM175. Na^+^/K^+^ exchange leads to capacitive currents that represent the binding and unbinding of the exchanged ions to the membrane surface [41]. These currents represent artifacts that overlay with the current resulting from channel conductivity measured on the TMEM175 sample. Artifact currents recorded with the SURFE^2^R N1 (Figure 1G) and the SURFE^2^R 96SE (Figure 1K) both show a negative amplitude of similar magnitude, indicating a tighter binding of Na^+^ to the membrane compared to K^+^, which is consistent with previous studies [41].

The time constants of the artifact currents are in the same range as the time constants derived from the currents recorded using the TMEM175 sample. Both ion binding to the lipid membrane and ion flux through a channel are fast processes, usually beyond the time resolution of the technique, which is limited by the solution exchange.

Sample dilution affects the size of the TMEM175 current (see above), but the higher dilution of control lysosomes does not reduce the average artifact current recorded with the SURFE^2^R 96SE, which is (−1.29 ± 0.52) nA compared to (−1.07 ± 0.3) nA for the SURFE^2^R N1, because ion–membrane interactions occur with both sample membranes adsorbed to the SSM and the SSM itself.

During assay development, the ratio between the artifact current and target protein current needs to be considered. To obtain currents that accurately represent ion translocation mediated solely by TMEM175, it is beneficial to subtract the currents recorded with control samples from those recorded with TMEM175 samples, specifically for high artifact to target protein current ratios.

### 2.2. Ion Selectivity of TMEM175 in an SSME Assay

To gain deeper insights into the properties of lysosomal TMEM175 and to validate the SSME technology, we performed SSME experiments to investigate cation selectivity and the dependence of the K^+^ flux rate on ion concentration.

#### 2.2.1. TMEM175 Conducts K^+^, Rb^+^, and Cs^+^, but Not Li^+^, Na^+^, or Choline^+^

To determine selectivity using a set of different cations, we performed 50 mM concentration jumps of Li^+^, Na^+^, K^+^, Rb^+^, Cs^+^, and choline^+^ against 50 mM NMDG^+^ in the non-activating solution using the SURFE^2^R N1 (Figure 2A). We performed the same experiment on lysosomes purified from HEK293 cells overexpressing TMEM175 (TMEM175 sample, Figure 2B) and control lysosomes that were purified from HEK293 cells expressing endogenous TMEM175 (control sample, Figure 2C), using a low sample dilution to enhance the signal-to-noise.

Comparing the average peak currents for different cations evoked in TMEM175 and control samples (Figure 2D), we detected similar translocation rates for K^+^, Rb^+^, and Cs^+^, but no significant conductivity for Na^+^, Li^+^, or choline^+^. Considering the resolution of the assay, the Na^+^ conductivity is at least two orders of magnitude lower compared to the conductivity for K^+^, Rb^+^, and Cs^+^.

We also performed a similar series of experiments on the SURFE^2^R 96SE, essentially reproducing the results found with the SURFE^2^R N1 (Figure 2E). Note that the Rb^+^ and Cs^+^ currents recorded with the SURFE^2^R 96SE seem to be more pronounced in both control and TMEM175 samples, compared to the SURFE^2^R N1, which may be attributed to different fluidic handling and sample concentrations. Appendix A contains representative current traces and additional experimental details.

Utilizing the HTS instrument, we employed 4-AP to examine the potential blockade of currents in both control and TMEM175 samples. The control sample remained unaffected by 4-AP, whereas the currents recorded from the TMEM175 sample were found to be inhibited by 4-AP. Note that the 4-AP inhibited currents recorded with the TMEM175 sample are identical to those obtained from the control sample. By subtracting the control sample currents from TMEM175 sample currents, we revealed that the TMEM175 net currents for K^+^, Rb^+^, and Cs^+^ were effectively inhibited by 10 mM 4-AP, resulting in complete blockade (Figure 2F).

**Figure 2 ijms-24-12788-f002:**
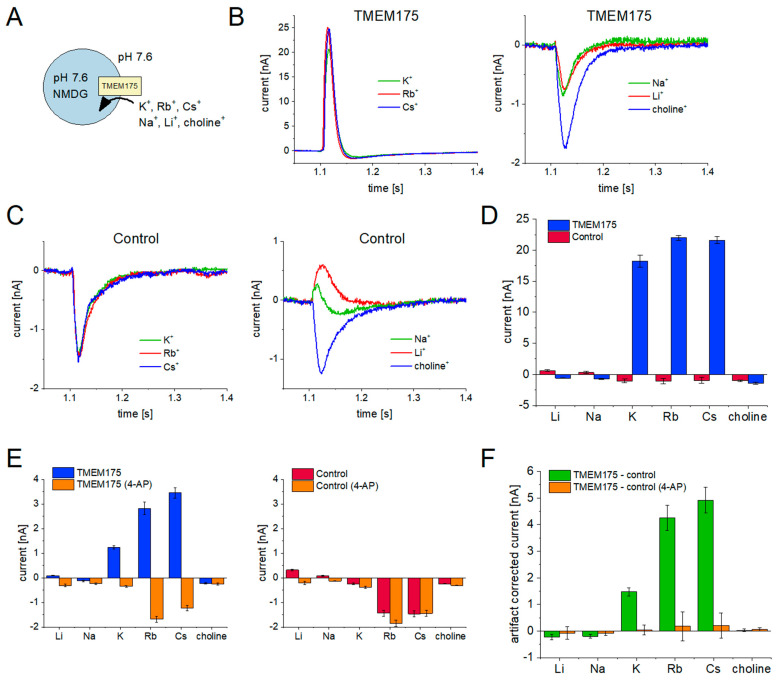
Cation selectivity of TMEM175 recorded with the SURFE^2^R N1 (**B**–**D**) and the SURFE^2^R 96SE (**E**,**F**): (**A**) Schematic of the experimental setting for the measurement of cation influx. We applied a solution exchange from 50 mM NMDG-Cl to 50 mM of a different Cl salt at pH 7.6; (**B**) Representative current traces recorded with the TMEM175 sample. All traces are recorded from the same sensor; (**C**) Representative current traces recorded with the control sample. Currents reflect background signals (artifacts) resulting from ion–membrane interactions; (**D**) Average peak currents and SD for N = 5 sensors recorded with the TMEM175 and control samples; (**E**) Average peak currents and SEM for N = 16 sensors, recorded with the TMEM175 (left) and control (right) sample. Blue and red bars represent the currents after application of 50 mM of the indicated cation during solution exchange, and orange bars represent the currents recorded with the same solution exchange in the presence of 10 mM 4-AP. Representative current traces are shown in Appendix A; (**F**) Processed data from (**E**). Control currents were subtracted from TMEM175 currents, both in the absence of 4-AP and in the presence of 4-AP. The currents reflect the TMEM175 net current.

#### 2.2.2. I/c Curves Reveal Kinetic Information about the Target Protein

In SSME recordings, voltage-clamp is not possible and current–voltage (I/V) relationships may not be determined as in conventional electrophysiology. Instead of testing the impact of the membrane electrical potential on the flux of the target protein, the impact of the chemical gradient is determined. In SSME experiments, varying ion concentration gradients are applied via solution exchange, which simultaneously act as a trigger for ion translocation through the target protein. Consequently, SSME generates current-concentration (I/c) curves as opposed to traditional I/V curves.

We applied K^+^ concentration jumps from 1 mM to 300 mM using the SURFE^2^R 96SE (Figure 3A), with a total of 12 different K^+^ concentrations and N = 8 sensors per concentration. K^+^ concentration jumps on lysosomes containing TMEM175 stimulated positive transient currents (Figure 3B), while control lysosomes showed negative transient currents (Figure 3C). Currents obtained from the TMEM175 sample increase with increasing K^+^ concentration in a hyperbolic manner (Figure 3D, blue curve), while the artifact currents obtained from the control sample exhibit a more linear dependence. Interestingly, the artifact currents start to be visible at higher K^+^ concentrations compared to currents recorded from the TMEM175 sample (inset of Figure 3D). The average current amplitude of the TMEM175 sample using 1 mM K^+^ is (166 ± 75) pA, while the average current amplitude of the control sample using 32 mM K^+^ is (−127 ± 17) pA. Currents below 50 pA are near the instrument’s noise level.

The artifact-corrected TMEM175-mediated net current was calculated by subtracting the average artifact current recorded with the control sample from the currents recorded with the TMEM175 sample (Figure 3E). The resulting I/c curve may be fitted using a hyperbolic equation (Figure 3E, green curve). Notably, the currents recorded using the four lowest (1–8 mM) and the four highest (120–300 mM) K^+^ concentrations each demonstrate linear relationships with different slopes, indicating that TMEM175 exhibits two distinct conductivity states depending on the available concentration of K^+^.

From the linear relationships, permeability coefficients may be obtained. According to Fick’s law, the passive flux J = P_K+_Δc linearly depends on the concentration difference across the membrane (Δc) and the permeability coefficient P for the respective ion. We found the potassium permeability at low potassium concentrations is increased four-fold (87 pA/mM) compared to the permeability at high concentrations (27.4 pA/mM).

**Figure 3 ijms-24-12788-f003:**
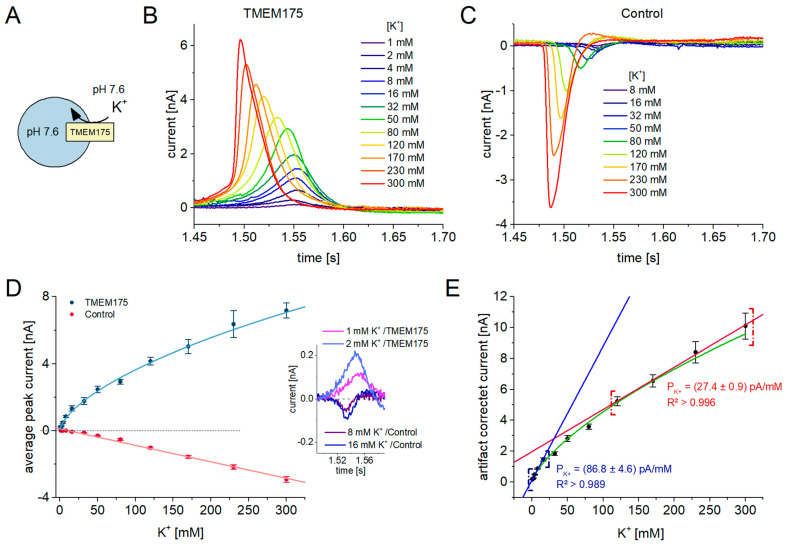
I/c curves for K^+^ flux through TMEM175 recorded with the SURFE^2^R 96SE: (**A**) Schematic of the experimental setting for the measurement of K^+^ influx. We applied K^+^ concentration jumps between 1 mM and 300 mM at pH 7.6 to measure K^+^ flux through TMEM175. Non-activating solution contained 300 mM NaCl, while activating solution contained different KCl concentrations (x) and 300 − x mM NaCl; (**B**) Representative current traces induced by K^+^ concentration jumps on lysosomes overexpressing TMEM175; (**C**) Representative current traces from the same experiment on control lysosomes. Currents reflect the capacitive charge displacement due to ion–membrane interactions on the surface of the sensor when the Na^+^/K^+^ solution exchange is applied. This resembles the background (artifact) current also underlying the currents recorded with the TMEM175 sample; (**D**) SEM and average peak currents recorded with control lysosomes (red) and lysosomes overexpressing TMEM175 (blue). Currents were recorded using the SURFE^2^R 96SE with N = 8 sensors per concentration. The inset shows current traces obtained from the control and TMEM175 samples when low K^+^ concentrations are used. Artifact currents are detected starting from 8 mM K^+^, while TMEM175 currents are already detectable starting from 1 mM K^+^; (**E**) SEM and average artifact-corrected peak currents reflecting the K^+^ flux through TMEM175, solely driven by the K^+^ concentration gradient. The concentration dependence appears to by hyperbolic (green curve); the first and last four datapoints were individually fitted using the linear regression I = P_K+_Δc with high fit quality, as indicated by the adjusted R² values. Two different permeability coefficients of P_K+_ for potassium were obtained, indicating that the conductivity state of TMEM175 depends on the K^+^ concentration.

### 2.3. Cytosolic pH, but Not pH Gradients Affect K^+^ Permeability of TMEM175

TMEM175 conducts both protons [6] and the larger monovalent cations K^+^, Rb^+^, and Cs^+^ [1,28]. A pH gradient directly affects the H^+^ translocation rate since it represents a driving force for H^+^ translocation. However, there is also evidence that H^+^ binding to TMEM175 controls a gating mechanism that changes the conductivity of TMEM175, as proposed by Hu et al. [6], which could also modulate the K^+^ flux through TMEM175.

To further investigate the effects of pH on K^+^ translocation through TMEM175, we performed 50 mM K^+^ concentration jumps at different pH values using the SURFE^2^R 96SE. We used two different settings: first, we employed identical pH values in the lysosomal lumen (pHi) and the external solution representing the cytosol (pHo). Subsequently, we applied different pHi values, keeping pHo constant at pH 7.5 to decipher the impact of lysosomal pH and pH gradients.

#### 2.3.1. K^+^ Conductivity Is Downregulated at Acidic and Alkaline pH

The initial set of experiments were carried out under symmetrical pH conditions, where the pH values inside and outside the lysosomes were identical (Figure 4A). First, we recorded the K^+^ flux through TMEM175 at pH 7.5 on each sensor using a 50 mM K^+^ concentration jump, which was used for normalization. We then proceeded to rinse the sensor using a non-activating solution with a different pH. Subsequently, each sensor was incubated for 5 min to ensure equilibration of the intra-lysosomal pH. Following this incubation period, the K^+^ concentration jump was repeated at the new pH to assess the rate of K^+^ translocation under these conditions. We repeated this measurement twice within 10 min to ensure proper pH equilibration and used the second data point for analysis.

We applied K^+^ concentration jumps using pH values ranging from pH 3.0 to pH 10.0, with a total of 12 different pH values and N = 8 sensors per pH. We performed the same experiment on the TMEM175 (Figure 4B, left) and control samples (Figure 4B, right). Artifact currents exhibit only minor pH dependence, while the pH dependent currents recorded with the TMEM175 sample display a bell-shaped activity curve (Figure 4C). From the normalized, artifact-corrected TMEM175 current, we were able to derive a pH optimum for TMEM175 activity and individual pK values for acidic and alkaline downregulation (Figure 4D). TMEM175 displays a pH optimum of 8.5. Alkaline downregulation occurs with pK = 10.1, while acidic downregulation requires two pK values to be accurately described: TMEM175 activity decreases between pH 8.0 and pH 6.0 with pK = 7.0 to about 30% of its maximum amplitude. Further reducing the pH will lead to a complete loss of K^+^ flux activity with pK = 4.5, potentially due to competition between recorded K^+^ and passive H^+^ fluxes. Using the SURFE^2^R N1, we were able to replicate these observations (Appendix A) and these findings are also in agreement with recent patch-clamp and structural investigations, which demonstrated that K^+^ flux through TMEM175 is downregulated under acidic pH conditions [6,42].

It may be counter-intuitive to observe the acidic downregulation of K^+^ flux through TMEM175 due to its native acidic environment. However, by using pH gradients, we were able to show that this effect is due to the acidification of the external pH, representing the cytosolic side of the membrane, which is tightly regulated in vivo and typically falls within the range of 7.0 to 7.5.

**Figure 4 ijms-24-12788-f004:**
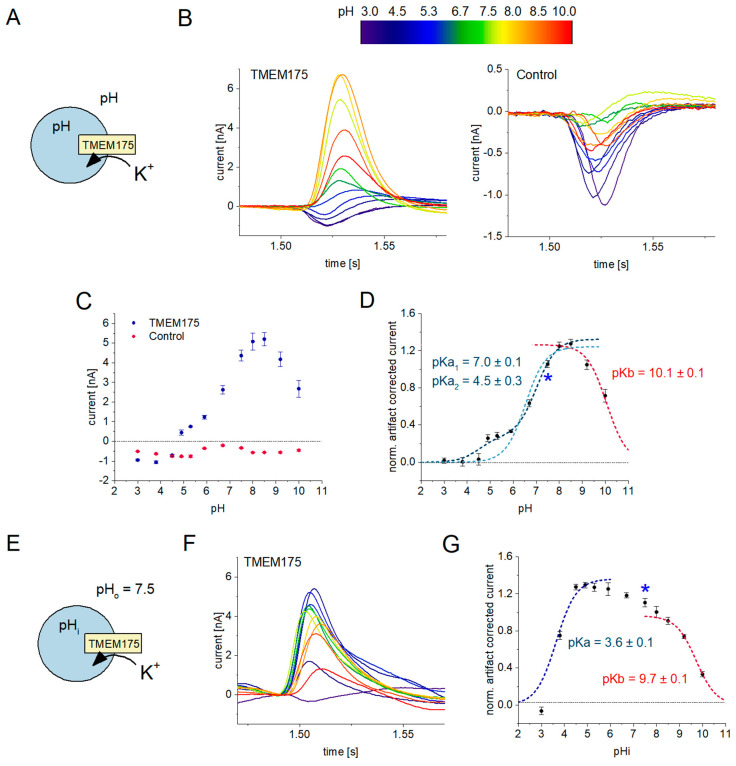
Effects of intra-lysosomal and cytosolic pH on K^+^ flux through TMEM175 recorded with the SURFE^2^R 96SE: the rainbow color scheme indicates the pH used for recording the currents shown in (**B**,**F**); (**A**) Schematic of the experimental setting for the measurement at symmetrical pH conditions, i.e., the cytosolic pH matches the intra-lysosomal pH. Sensors were prepared at pH 7.6 and then rinsed with non-activating solution, which was adjusted to the desired pH, followed by an incubation of 5 min for pH equilibration. The non-activating solution contained 50 mM NaCl. We then performed a solution exchange to the activating solution, exchanging 50 mM Na^+^ with 50 mM K^+^ to measure K^+^ flux through TMEM175 at the given pH; (**B**) Representative current traces recorded at different pH values on lysosomes overexpressing TMEM175 (left) and control lysosomes (right); (**C**) SEM and average peak currents from N = 8 sensors per pH recorded with control lysosomes (red) and lysosomes overexpressing TMEM175 (blue); (**D**) SEM and average, artifact-corrected, normalized peak currents reflecting the K^+^ flux through TMEM175 at the given pH, solely driven by the K^+^ concentration gradient. We applied individual fits using titration equations to derive pK values for acidic and alkaline downregulation. For acidic downregulation, we used the fitting equation I = I_max_/(1 + 10^pKa-pH^) with a single pK, achieving a poor fit of the data with pKa = 6.5 ± 0.1 (light blue dashed line). We then applied the fitting equation I = I_max1_/(1 + 10^pKa1-pH^) + I_max2_/(1 + 10^pKa2-pH^) with two pK values (dark blue dashed line), achieving a good fit, and considering the activity plateau observed between pH 6 and pH 5. Alkaline downregulation is described via a single pKb (red dashed line), using the equation I = I_max_/(1 + 10^pH-pKb^). The blue asterisk indicates the pH 7.5 condition, which is identical across the datasets with symmetrical pH (this graph) and pH gradients across the lysosomal membrane (graph in (**G**)); (**E**) Schematic of the experimental setting for the measurement using pH gradients across the lysosomal membranes. Sensors were prepared at pH 7.6 and then rinsed with non-activating solution set to the desired lysosomal pH (pHi), followed by an incubation of 10 min for pH equilibration. The non-activating solution contained 50 mM NaCl. We then performed a solution exchange to the non-activating solution set to pH 7.5, which defines the external pH (pHo). Less than 300 ms after exchanging the external pH, the activating solution is applied, exchanging 50 mM Na^+^ with 50 mM K^+^ to measure K^+^ flux through TMEM175 in the presence of the pH gradient; (**F**) Representative current traces recorded in the presence of different pH gradients on lysosomes overexpressing TMEM175. The color of the trace reflects the intra-lysosomal pH according to the scheme in (**B**); (**G**) SEM and average, artifact-corrected, normalized peak currents reflecting the K^+^ flux through TMEM175 at the given intra-lysosomal pH (pHi), keeping the external pH constant (pHo = 7.5). Acidic and alkaline downregulations were fitted using single pK equations as explained in (**D**). Between pHi 9 and pHi 4, TMEM175 activity increases in a linear fashion.

#### 2.3.2. Intra-Lysosomal Protons Do Not Affect the K^+^ Flux through TMEM175

From the pH dependence described above, it is not clear whether cytosolic pH (pHo), lysosomal pH (pHi), or both are causing the downregulation of K^+^ flux through TMEM175 at acidic and alkaline pH values. Because TMEM175 is exposed to a pH gradient in its natural environment, it is likely that lysosomal and cytosolic acidification have different effects on the channels’ activity. To investigate and distinguish the effects of lysosomal and cytosolic pH individually, we applied different pH gradients before measuring K^+^ translocation through TMEM175 (Figure 4E).

After the measurement of the K^+^ flux through TMEM175 at pH 7.5, sensors are rinsed with non-activating solution at a certain pH, following incubation for 10 min to equilibrate the lysosomal pH. But instead of a measurement at the new pH (as in the experiments described above), a pH gradient is established with the beginning of the next measurement sequence: non-activating solution at pH 7.5 is flushed across the sensor and defines the cytosolic pH, which is the same for all measurements. About 0.3 s after adding non-activating solution at pH 7.5, the activating solution at pH 7.5 reaches the sensor, and a 50 mM K^+^ concentration jump stimulates K^+^ translocation through TMEM175 in the presence of the pH gradient. The key for these types of experiments is the stability of pH gradients. They are not stable on the time scale of 10 min, allowing to set the lysosomal pH via incubation, but the pH gradients were stable on the time scale of hundreds of milliseconds required for the measurement.

We performed this assay on TMEM175 and control samples using the same pHi values and plate layout, as for the experiment using symmetrical pH values. Representative TMEM175 sample current traces are shown in Figure 4F. The pHi dependence of normalized, artifact-corrected TMEM175 net currents are shown in Figure 4G. In the presence of a pH gradient, the current amplitudes recorded with the control sample do not exhibit significant differences compared to symmetrical pH conditions. However, the pH dependency of the TMEM175 sample current undergoes substantial changes when only the intra-lysosomal pH is acidified, in contrast to the acidification of both the intra-lysosomal and cytosolic pH.

Similar to the results obtained under symmetrical pH conditions, we observed a downregulation of K^+^ flux through TMEM175 at increasingly alkaline or acidic intra-lysosomal pH values, with pK values of 9.7 and 3.6, respectively (Figure 4G). However, when the pHi is varied between 4.5 and 8.5, with the significant downregulation occurring in the absence of a pH gradient (Figure 4D), a broad plateau with high TMEM175 activity is observed. We conclude that the downregulation of K^+^ flux through TMEM175 observed under symmetrical pH conditions between pH 8.5 and pH 4.5 is a consequence of cytosolic pH.

Interestingly, we observed that the K^+^ flux through TMEM175 increased by ~20% when lysosomal pH is acidified from pH 8.5 to pH 4.5 (Figure 4G), which may indicate H^+^ activation of TMEM175 from the lysosomal side of the membrane. However, this increase is linear with pH, which does not fit to the binding of a single H^+^ to TMEM175, leading to a different conductivity state. A linear dependence may be the result of an increasingly negative membrane potential that is generated as a consequence of the H^+^ efflux across the lysosomal membrane when the pH gradient is applied.

It must be noted that TMEM175 activity exhibits a sharp decline below an intra-lysosomal pH of 4.5, with a pK value of 3.6 (Figure 4G). We propose that this downregulation, like the pK = 4.5 downregulation under symmetrical pH conditions, is a result of competition between H^+^ and K^+^ fluxes.

### 2.4. H^+^ Permeability of TMEM175 in an SSME Assay

A recent report has indicated that the physiological function of TMEM175 represents the conduction of H^+^ instead of K^+^ [6]. Specifically, it has been demonstrated that TMEM175 functions as a proton-gated H^+^ channel. The authors observed that TMEM175 promotes the efflux of protons when the pH of lysosomes falls below pH 4.5. Conversely, they found that the conduction of H^+^ is reduced as the pH of lysosomes rises above pH 5.0. The authors suggest that TMEM175 plays a crucial role in maintaining the pH of lysosomes within the range of pH 4.5 to pH 5.0. This regulation occurs in collaboration with the v-type H^+^-ATPase, which is responsible for the uptake of H^+^.

#### 2.4.1. H^+^ Translocation in Influx Mode

To further investigate the capability of TMEM175 to translocate protons using SSME, we performed H^+^ concentration or pH jumps using the SURFE^2^R N1. The non-activating solution was set to pH 7.6, defining both the lysosomal and cytosolic pH before the start of the experiment. By exchanging the non-activating for the activating solution, we acidified the cytosolic solution and stimulated H^+^ influx through TMEM175 (Figure 5A). We used ΔpH values between 0.2 and 3.0 for multiple measurements on the same sensor and found that H^+^ currents increase with ΔpH (Figure 5B).

However, the application of pH jumps lead to significant artifact currents of positive magnitude, as observed with the control lysosomes (Figure 5C). These artifacts are a result of protons interacting with the lipid membranes and/or unknown H^+^ leak pathways through the lysosomal membrane. To conclude pure TMEM175-related H^+^ currents, we subtracted the artifact currents from the currents recorded with the TMEM175 sample. When plotting ΔpH against the artifact-corrected average TMEM175 net H^+^ current, we obtain a close-to-linear correlation (Figure 5D). We also used the TMEM175 inhibitor 4-AP to validate these results: while 4-AP did not affect the current recorded with the control lysosomes, the current recorded from the TMEM175 sample was reduced to the level of the control current, indicating a full block (black curves in Figure 5B,C).

For comparison with the I/c curve reported for K^+^ (Figure 3E), we calculated the difference in H^+^ concentration across the lysosomal membrane Δc = c_H_^out^ − c_H_^in^, which correlates linearly with the passive H^+^ flux J = P_H+_Δc according to Fick’s law. We observed a hyperbolic correlation between the recorded H^+^ currents and Δc and determined an EC_50_ of 9.5 µM (Figure 5E), corresponding to ~pH 5. Hence, H^+^ concentration jumps to ~pH 5 will half-saturate H^+^ influx, which likely is a consequence of acidic downregulation upon cytosolic acidification, as outlined above (Figure 4D).

**Figure 5 ijms-24-12788-f005:**
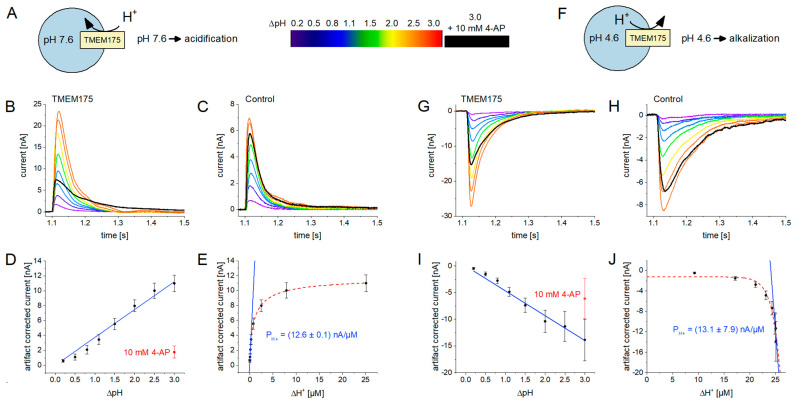
H^+^ currents through TMEM175 recorded with the SURFE^2^R N1: the figure shows results for H^+^ influx (**A**–**E**) and efflux (**F**–**J**) experiments; the rainbow color scheme indicates the ΔpH used to stimulate the currents shown in (**B**,**C**,**G**,**H**); (**A**) Schematic of the experimental setting for the measurement of H^+^ influx. We used H^+^ concentration or pH jumps to measure H^+^ flux through TMEM175. Non-activating solution was set to pH 7.6, while activating solution contained different pH values between pH 7.4 and pH 4.6; (**B**) Representative current traces induced by pH jumps on lysosomes overexpressing TMEM175. All current traces were recorded with the same sensor, starting from the lowest ΔpH. After recording using ΔpH = 3.0, the sensor was incubated with 10 mM 4-AP for 3 min, followed by the repetition of the solution exchange with ΔpH = 3.0 in the presence of 10 mM 4-AP (black trace). The remaining current mostly reflects the background (artifact) current after full block of TMEM175; (**C**) Representative current traces from the same experiment on control lysosomes. Currents reflect the capacitive charge displacement due to ion–membrane interactions on the surface of the sensor when the pH jump is applied. This represents the artifact current also underlying the currents recorded with the TMEM175 sample; (**D**) SEM and average, artifact-corrected, normalized peak currents from N = 4 sensors, reflecting the H^+^ flux through TMEM175, solely driven by the pH gradient. The current linearly depends on ΔpH; (**E**) The same data as shown in (**D**), re-plotted to obtain an I/c curve visualizing the dependence of the H^+^ flux through TMEM175 on H^+^ concentration. Only the first three data points show a linear dependence. Fitting using the linear regressions I = P_H+_Δc reveals the H^+^ permeability of TMEM175 in influx mode. The activity plateau when higher H^+^ concentrations are used likely corresponds to the inhibitory effect of cytosolic acidification (Figure 4D); (**F**) Schematic of the experimental setting for the measurement of H^+^ efflux. Non-activating solution was set to pH 4.6, while activating solution contained different pH values between pH 4.8 and pH 7.6; (**G**–**J**) Same plots and fits as described in (**B**–**E**), but for the H^+^ efflux experiment described in (**F**). In (**J**), the last three data points were used for the linear fit to obtain the H^+^ permeability of TMEM175 in efflux mode; here, large H^+^ concentration jumps are required to achieve close-to-neutral cytosolic pH values that prevent downregulation of TMEM175 due cytosolic acidification (Figure 4D).

#### 2.4.2. H^+^ Translocation in Efflux Mode

To further test the hypothesis of TMEM175 being H^+^ gated, we reversed the assay conditions. Using an acidic non-activating solution, we set the lysosomal and cytosolic pH to 4.6 and then performed pH jumps to alkalize the external medium to drive H^+^ efflux through TMEM175 (Figure 5F). We selected the same ΔpH values as in the influx experiment. The major difference is the initially acidic intra-lysosomal pH that was shown to increase the conductivity of TMEM175 [6]. The results are very similar to the influx experiment: the TMEM175 sample reveals large currents of negative amplitude representing H^+^ efflux (Figure 5G) and amplitudes increasing with ΔpH. As expected, the control sample reveals artifact currents (Figure 5H) and the artifact-corrected average TMEM175 H^+^ current shows a linear correlation with ΔpH (Figure 5I). H^+^ efflux currents were also inhibited using 4-AP, indicating that the currents originate from H^+^ flux through TMEM175; however, the potency of inhibition was lower compared to the inhibition of H^+^ influx.

We also plotted the recorded H^+^ currents against the H^+^ concentration difference across the lysosomal membrane to obtain an I/c curve and found that the H^+^ currents increase exponentially with Δc (Figure 5J). Again, this is likely a result of the acidic downregulation of TMEM175 when the cytosol acidifies (Figure 4D). Only larger H^+^ concentration jumps reaching pH values close to the physiological cytosolic pH will allow TMEM175 to catalyze efficient H^+^ efflux.

When we compare H^+^ fluxes in the presence of inward and outward directed pH gradients of ΔpH 3.0 over the lysosomal membrane, the H^+^ efflux rate is 26% larger compared to the H^+^ influx rate (Figure 5E,J), which may be interpreted as H^+^ activation due to lysosomal acidification. But again, this seems to be a result of changes in cytosolic pH that affect TMEM175 conductivity, which is also observed in our K^+^ assay (Figure 4D).

#### 2.4.3. A Quantitative Comparison between Permeabilities

To address the questions regarding whether TMEM175 conducts H^+^ rather than K^+^ and whether TMEM175 exhibits higher H^+^ permeability when the lysosomal lumen is acidified, we compared the permeability coefficients for H^+^ and K^+^ fluxes.

Since TMEM175 currents are modulated by cytosolic pH changes (Figure 4D), we do not observe a linear I/c relationship when different H^+^ concentrations are applied as a driving force (Figure 5E,J). However, we can derive a linear relationship when analyzing only small cytosolic pH changes close to the physiological cytosolic pH for which the TMEM175 activity is stable. To do so, we used a linear fit of the first three datapoints of the influx dataset (pH_i/o_ 7.6 -> pH_o_ 7.4/7.2/6.6) and the last three data points of the efflux dataset (pH_i/o_ 4.6 -> pH_o_ 7.6/7.1/6.6) (Figure 5E,J).

We obtained permeabilities of (12.6 ± 0.1) nA/µM for H^+^ influx when the lysosomal pH is 7.6 and (13.1 ± 7.9) nA/µM for H^+^ efflux when the lysosomal pH is 4.6. These values are very similar, suggesting that we do not observe H^+^ gating when the lysosomal lumen is acidified. However, the standard error of the permeability obtained in efflux mode when lysosomal pH is 4.6 is large and potential H^+^ gating below pH 4.5 cannot be excluded.

Comparing the permeability for H^+^ of 13 nA/µM with that found for K^+^ (27 and 87 pA/mM, Figure 3E), it becomes evident that the permeability for H^+^ is several orders of magnitude larger compared to the K^+^ permeability. H^+^ and K^+^ assays have been performed with different instrumentation using different TMEM175 sample amounts. To ensure an accurate comparison of the determined permeabilities, we normalized the permeability coefficients to the average current values obtained from 50 mM K^+^ concentration jumps on the two different devices with different sample amounts: (2.8 ± 0.2) nA for the SURFE^2^R 96SE and (18.2 ± 0.9) nA for the SURFE^2^R N1. We conclude a permeability ratio of P_H_/P_K_ = 74,000 for the physiological K^+^ concentration range and P_H_/P_K_ = 23,000 for K^+^ concentrations < 50 mM, which is consistent with literature findings, i.e., P_H_/P_K_ > 100,000 [42], and fits well with the ratio found using the whole-cell patch-clamp of P_H_/P_K_ = 48,000 [6].

This analysis also illustrates that the chemical gradients applied in SSME and the voltage steps used in patch-clamp recordings are equivalent driving forces and both may be used to conclude information about ion permeabilities. Additionally, the different read-outs—steady-state currents in patch-clamp and capacitive currents in SSME—do not affect these results.

### 2.5. Tool Compounds Affect TMEM175 Activity in an HTS-Compatible SSME Assay

The SURFE^2^R 96SE was developed to serve as a device for secondary compound screening efforts, enabling the hit validation and determination of IC_50_ or EC_50_ values with a reasonable throughput of up to 96 parallel measurements. The high level of parallelization and the ability to conduct sequential measurements offer a considerable amount of experimental flexibility. We tested several possible experimental settings and established a procedure for compound testing, which represents a compromise between throughput, sufficient repetitions for statistical analysis, and control experiments, which is described in detail in Appendix A.

Although it is generally feasible to examine the impact of compounds on H^+^ flux through TMEM175, we opted to utilize the K^+^ flux assay (Figure 6A). This decision was based on two factors: first, the H^+^ flux assay tends to generate higher levels of artifact currents, resulting in reduced accuracy (Figure 5C,H); second, the pH alterations used to stimulate H^+^ flux through TMEM175 also influence the conductivity state of TMEM175 (Figure 4D and Figure 5E,J), making assay design and data interpretation more challenging.

In brief, compound measurements using SSME are performed in a three-step process (Figure 6B). First, TMEM175-mediated K^+^ translocation is measured through 50 mM K^+^ concentration jumps. This represents the baseline current of TMEM175 activity (I_0_). Second, the compound is added at the desired concentration, following an incubation time of 5–15 min to equilibrate hydrophobic compounds across the membranes on the sensor. Third, the activation through a 50 mM K^+^ concentration jump is repeated, this time in the presence of the compound to record the modulated TMEM175 current (I). In this process, each individual sensor of the 96-well sensor plate is only exposed to a single concentration of the compound. The relative current amplitude I/I_0_ is used as a measure of the compound effect and averaged across sensors.

To evaluate the quality of the assay, we measured a total of ten 96-well sensor plates to determine the z’ prime and success rates (Figure 6C). For the calculation of z’ prime, we used a 50 mM K^+^ concentration jump as a negative control and the same concentration jump in the presence of 10 mM 4-AP as the positive control, essentially achieving a full block. The determined z’ prime of 0.87 ± 0.027 indicates an excellent assay quality [43]. Using the QC criteria outlined in the Methods section, we achieved a superior success rate of 96.4 ± 3.3%.

**Figure 6 ijms-24-12788-f006:**
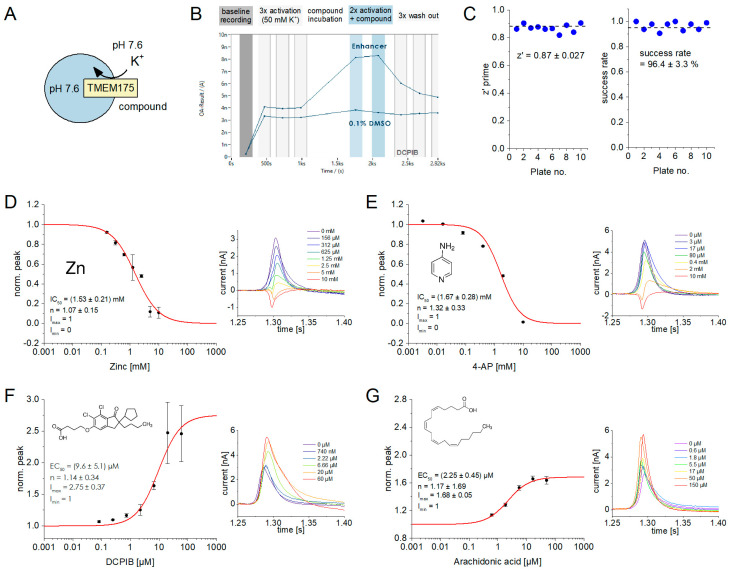
Effects of inhibitors and enhancers on K^+^ flux through TMEM175 recorded with the SURFE^2^R 96SE: (**A**) Schematic of the experimental setting to measure the effects of compounds on TMEM175. We applied 50 mM K^+^ concentration jumps in the absence and presence of different compound concentrations to derive information about compound effects on K^+^ flux through TMEM175; (**B**) Time dependence of recorded peak current amplitudes for the complete workflow, which involves multiple sequential measurements. Two representative sensor wells are shown: one sensor was exposed to 50 µM DCPIB, while the other sensor was treated with 0.1% DMSO (negative control). The recording of each sensor well starts with the application of a control measurement, rinsing non-activating solution across the sensor (baseline recording). Afterwards, we performed three solution exchange measurements from 50 mM NaCl to 50 mM KCl to measure K^+^ flux through TMEM175 (baseline TMEM175 current, I_0_). We then rinsed the sensor with non-activating solution containing the compound at a given concentration, followed by incubation of 15 min to equilibrate the sensors with the compound. Subsequently, the solution exchange from 50 mM NaCl to 50 mM KCl is repeated twice, but in the presence of the given compound concentration to determine the modulated peak current I. Afterwards, we repeated the measurement in the absence of the compound three times (wash-out) to recover the non-modulated TMEM175 current. We subtracted the average artifact current recorded with control lysosomes from I and I_0_ (Figure 1K), without considering off-target effects of the tested compound. We then performed in-well normalization of artifact-corrected currents (I/I_0_) followed by averaging across sensors, before plotting the IC_50_ and EC_50_ curves shown in (**D**–**G**); (**C**) Z’ prime values and success rates from ten 96-well sensor plates. Success rates were determined based on the QC criteria provided in the Methods section. Z’ prime values were established based on the currents recorded after a solution exchange from 50 mM NaCl against 50 mM KCl (negative control) and the same solution exchange experiment in the presence of 10 mM 4-AP (full block, positive control). We determined average values, AVG_pos_ and AVG_neg_, and standard deviations, SD_pos_ and SD_neg_, of the recorded peak currents. For the calculation, we used the equation z’ = [(AVG_po_ − 3SD_pos_/√N) − (AVG_neg_ − 3SD_neg_/√N)]/(AVG_pos_ − AVG_neg_), which considers the number of datapoints N [44]; (**D**–**G**) IC_50_ and EC_50_ curves (left) and representative current traces recorded in the presence of different compound concentrations (right) for ZnCl_2_ (**D**), 4-AP (**E**), DCPIB (**F**), and arachidonic acid (**G**). Error bars represent the SEM obtained from automated data analysis using DataControl96, based on variations of the currents obtained with the TMEM175 sample. Each compound concentration was recorded with at least N = 8 sensors. The equation I = I_max_ − (I_max_ − I_min_)/(1 + (c/EC_50_)^n^) was used to fit all datasets. I_min_ was always fixed to 0 for inhibitors or 1 for enhancers, as indicated.

#### 2.5.1. 4-Aminopyridine and Zinc Act as Inhibitors

4-AP and zinc have been described as pore blockers that reduce the conductivity of TMEM175 [1,45]. We have used concentrations between 3 µM and 10 mM to establish IC_50_ values for these inhibitors. For zinc and 4-AP, we found IC_50_ values of (1.53 ± 0.21) mM (Figure 6D) and (1.67 ± 0.28) mM (Figure 6E), respectively. These values are higher than the IC_50_ values reported in the literature, which range from 30 µM to 60 µM for both inhibitors [1,45]. In our SSME experiment, 10 mM of 4-AP or zinc were sufficient for a full block of TMEM175.

#### 2.5.2. DCPIB and Arachidonic Acid Stimulate TMEM175 Activity

DCPIB and arachidonic acid have been described to enhance the K^+^ and H^+^ conductivity of TMEM175 [6]. We used concentrations between 60 nM and 60 µM to establish the EC_50_ values for these compounds. For DCPIB and arachidonic acid, we determined EC_50_ values of (9.6 ± 5.1) µM (Figure 6F) and (2.3 ± 0.5) µM (Figure 6G), respectively. Compared to the TMEM175 baseline current (100%), the maximum potentiation was 275 ± 37% for DCPIB and 168 ± 5% for arachidonic acid.

### 2.6. Whole-Cell Automated Patch-Clamp Recordings

Section 2.1, Section 2.2, Section 2.3, Section 2.4 and Section 2.5 showcase the results obtained from SSME recordings utilizing purified lysosomes from HEK293 cells overexpressing TMEM175. In this section, we present results from the same cell line, but employ whole-cell APC recordings. This approach is feasible since the overexpression of endosomal membrane proteins can lead to their accumulation on the plasma membrane [46], a phenomenon demonstrated previously for TMEM175 [3,28].

#### 2.6.1. Establishing an APC Assay for TMEM175

In our APC recordings, we observed outward currents, which correspond to the flux of Cs^+^ through TMEM175 (Figure 7A,B). Due to reversed TMEM175 orientation in the plasma membrane, this physiologically corresponds to lysosomal K^+^ influx driven by the K^+^ concentration gradient, a similar condition as used in the SSME-based assay (Figure 6A).

To validate the APC assay, we compared currents between TMEM175-overexpressing cells and control HEK293 UT cells. The average baseline current amplitudes recorded with the TMEM175-overexpressing cells are (3.87 ± 2.93) nA and are more than 10 times larger compared to the K^+^ currents recorded from HEK293 plasma membrane in previous studies [2,3]. Using control cells, the average baseline current amplitudes were (0.22 ± 1.72) nA. In our compound application protocol, which was used to evaluate compound potencies, we consistently administered tool compounds to assess their impact on TMEM175 activity (Figure 7C). We used 0.5 mM ZnCl_2_ to enhance the TMEM175 current, followed by a full block using 1 mM 4-AP. Under these conditions, we obtained a signal enhancement of 301 ± 94% and a signal block of 73 ± 14% over 125 plates, respectively.

We also validated the robustness of the established APC assay using datasets recorded from 125 individual 384-well sensor plates. Based on the strict QC criteria outlined in the Methods section, we achieved a success rate of 82 ± 5% (Figure 7D). We determined a z’ prime value of 0.77 ± 0.06 when comparing a full block using 1 mM 4-AP as a positive control with the current upon addition of 0.2% DMSO as a negative control (Figure 7D), which attests to the excellent assay conditions for testing compounds [43].

**Figure 7 ijms-24-12788-f007:**
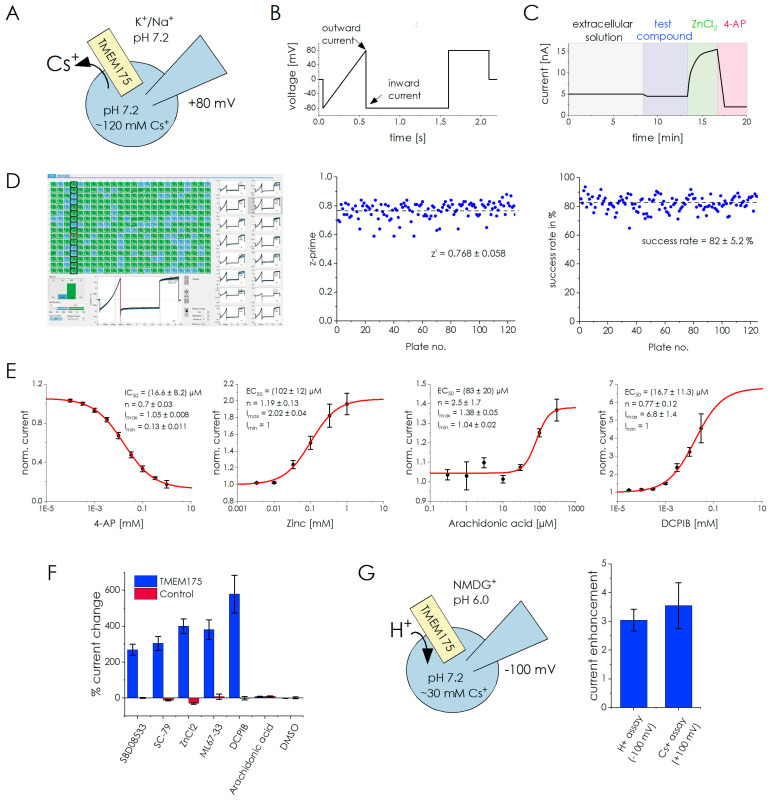
Whole-cell APC recordings on TMEM175: (**A**) Schematic representation of assay conditions in the Cs^+^ flux assay. The ions known to permeate through TMEM175 in the external and internal solutions are indicated; (**B**) Voltage protocol and time points for the read-out of inward and outward currents; (**C**) The compound application protocol consists of the recording of the TMEM175 baseline current. Subsequently, currents are recorded following the addition of a variable test compound, a tool enhancer (0.5 mM zinc), and a tool inhibitor (1 mM 4-AP); (**D**) Screenshot of DataControl384 used for automated analysis of 384 parallel recordings (left) and obtained success rates and z’ prime values over 125 individual 384-well sensor plates (right). Success rates were determined based on the QC criteria provided in the Methods section. Z’ prime values were established using 0.2% DMSO (negative control) and 1 mM 4-AP (positive control). We determined average values, AVG_pos_ and AVG_neg_, and standard deviations, SD_pos_ and SD_neg_, of the recorded currents. The calculation we used was equation z’ = [(AVG_pos_ − 3SD_pos_/√N) − (AVG_neg_ − 3SD_neg_/√N)]/(AVG_pos_ − AVG_neg_), which considers the number of datapoints N [44]; (**E**) Dose–response curves for the tool compounds 4-AP, zinc, arachidonic acid, and DCPIB, applying the Cs^+^ flux assay. Datapoints were normalized to the TMEM175 baseline current before compound addition and then averaged across at least N = 5 cells. Error bars represent the SEM obtained from automated data analysis using DataControl384. The equation I = I_max_ − (I_max_ − I_min_)/(1 + (c/EC_50_)^n^) was used to fit all datasets; (**F**) relative current changes after application of 50 µM of different enhancers to TMEM175 cells (blue) or control HEK293 UT cells expressing endogenous TMEM175 (red), normalized to the baseline current (0%). In the case of DMSO, 0.1% (*v*/*v*) DMSO was used; (**G**) Schematic representation of assay conditions in the H^+^ flux assay (left) and comparison of current potentiation after application of 30 µM DCPIB in the Cs^+^ flux and H^+^ flux assays (right).

#### 2.6.2. Investigating Compound Potencies in Whole-Cell APC

We have tested the effects of 4-AP, zinc, DCPIB, and arachidonic acid on TMEM175 channel activity (Figure 7E). For the pore blocker 4-AP, we found an IC_50_ of (16.6 ± 8.2) µM, matching with literature values [1,28,45]. Zinc has been also described as a pore blocker with similar potency as 4-AP [1]. However, in our assay, zinc induced signal enhancement with an EC_50_ of (102 ± 12) µM. To exclude potential inhibitory effects at different concentration ranges, we used concentrations of up to 10 mM without further impacting the recorded currents.

DCPIB and arachidonic acid have been described as enhancers of TMEM175 activity [6]. For DCPIB, we determined EC_50_ = (16.7 ± 11.3) µM and a (6.8 ± 1.4)-fold maximum potentiation. For arachidonic acid, we obtained EC_50_ = (83 ± 20) µM and a 34% maximum potentiation using our standard assay that contained 0.05% BSA. Since it was found that BSA binds arachidonic acid [47], we repeated the assay in the absence of BSA and found that the EC_50_ and maximum potentiation were unaltered. The tool compound data obtained from whole-cell APC and SSME recordings are summarized in Table 3.

#### 2.6.3. Off-Target Compound Effects in Whole-Cell Automated Patch-Clamp Are Negligible

To evaluate potential off-target compound effects, we compared the current potentiation of a set of compounds at 50 µM concentration between TMEM175 cells and control cells (Figure 7F). The compound effects on control cells were between +10% and -40% compared to the control baseline current. ZnCl_2_ showed the largest effects and decreased the baseline current by 30–40%. In contrast, the tested compounds increased the currents recorded with TMEM175 cells between 270% and 580%. Therefore, off-target compound effects seem to be negligible in the APC assay.

#### 2.6.4. The Potency of DCPIB Is Identical for APC-Based Cs^+^ Flux and H^+^ Flux Assays

Since the physiological role of TMEM175 involves H^+^ flux and compounds may have different impacts on H^+^ and Cs^+^ fluxes, we investigated the effect of DCPIB in an APC-based H^+^ flux assay (Figure 7G). The recorded inward currents physiologically correspond to lysosomal H^+^ efflux driven by the pH gradient, due to the reversed TMEM175 orientation in the plasma membrane.

We found that the inward current of TMEM175 cells increases 60-fold when external pH is acidified to the native lysosomal pH range. Average H^+^ currents under the given conditions are ~20 times larger compared to the average currents recorded in our Cs^+^ flux assay, suggesting a high selectivity of TMEM175 for H^+^. The inward current recorded from control HEK293 UT cells is only slightly affected by external acidification, possibly corresponding to the activity of pH-sensitive chloride channels.

We tested the effect of 30 µM DCPIB on TMEM175 in our H^+^ assay at pH 6.0 (Figure 7G). The inward current at −100 mV increased by 304 ± 38%. This corresponds to a similar effect compared to the effect in the Cs^+^ flux assay, which was 355 ± 80% at +100 mV.

### 2.7. Assay Technologies Affect Apparent Drug Potencies—A Case Study on Nine Enhancers Using Lysosomal Patch-Clamp, Whole-Cell APC and SSME

We observed significant discrepancies in the effects of the tool compounds when comparing the whole-cell APC recordings and SSME recordings: zinc acts as an enhancer in APC, but as an inhibitor in SSME; arachidonic acid is 40-fold more potent in SSME recordings, while 4-AP is 100-fold less potent in SSME recordings compared to APC (Table 3).

Since both technologies apply different assay conditions and samples, the origin of the discrepancies may not be concluded. To investigate potential systematic variations among different assay technologies, namely, whole-cell APC, SSME on purified lysosomes, and manual LPC, we conducted a comparative analysis of the potency and maximum potentiation of a set of eight blinded compounds and SC-79 (Figure 8, Table 4). SSME recordings were performed using the same procedures as described for the tool compounds above. However, adjustments were made to the voltage application protocol in APC, as outlined in the Methods section. Lysosomal patch-clamp was performed using a manual setup. In contrast to APC and SSME recordings, the recorded current in LPC represents H^+^ efflux from lysosomes in the presence of a physiological pH gradient (details are provided in the Methods section).

#### 2.7.1. SC-79 Has No Significant Effect on TMEM175 in SSME Due to the Absence of PKB

It has been shown previously that the addition of SC-79 to patch-clamped lysosomes leads to increasing TMEM175 ion conductivity via protein kinase B (PKB) activation [48]. A different report did not find any effect of SC-79 on TMEM175 in whole-cell patch-clamp recordings [6].

Using whole-cell APC, we found an EC_50_ of (62 ± 43) µM and a current potentiation of 245 ± 102% (Figure 8A, Table 4), supporting the findings by Wie et al. With the SSME assay, we observed only a small potentiation of 9 ± 4%. From the dose–response curve, an EC_50_ value of (2.9 ± 3.1) µM could still be determined. However, measurements on control lysosomes indicate that this apparent potentiation is due to off-target effects (see below, Figure 9F). During lysosomal purification for SSME recordings, the soluble contents of the cytoplasm were removed and, unfortunately, the TMEM175-PKB complex did not survive this procedure, hindering the activation of TMEM175 via SC-79. Interestingly, with LPC, we find an EC_50_ ((21 ± 32) µM) and potentiation (91 ± 52%) that are similar to the results obtained with APC (Figure 8A, Table 4). Evidently, the TMEM175-PKB complex survives the purification and vacuolin-1 treatment of the lysosomes for LPC experiments, highlighting an advantage of utilizing freshly purified lysosomes.

#### 2.7.2. We Grouped Test Compounds into Three Clusters Based on Their EC_50_ Values

Similar to the tested tool compounds (Figure 6 and Figure 7), the EC_50_ values determined for some compounds differ between methods (Table 4). It should be noted that the set of compounds were not ideal due to the overall low potency in the µM range, with 100 µM being the highest compound concentration used. In some cases, the highest compound concentration used did not saturate the response, hindering the determination of EC_50_ and I_max_ values.

Based on the EC_50_ values determined, we grouped the eight compounds into three clusters: two compounds generated very similar EC_50_ values across all three assay technologies (cluster 1, Figure 8B), three compounds demonstrated no activity in LPC, but higher activities with the other two techniques (cluster 2, Figure 8C), and three compounds showed moderate to high affinity in LPC, but lower activities with the other two techniques (cluster 3, Figure 8D). Comparing the data of cluster 2 and cluster 3, the apparent compound potency recorded with SSME falls between those obtained with whole-cell APC and LPC. This observation suggests that the discrepancy in compound potency may arise from the characteristics of the lysosomal membrane, because lysosomal samples used for SSME recordings are contaminated to some degree with plasma membrane vesicles (Figure 1B).

Cluster 2 contains compounds that can be interpreted as false positives in recordings using TMEM175 from the plasma membrane. On the other hand, cluster 3 comprises compounds that can be interpreted as false negatives in such recordings. However, additional studies are warranted, as the current results are unable to rule out false positives for LPC, especially for compounds that show similar results in APC and SSME, i.e., compounds **6** and **7** within cluster 3 (Figure 8C). In this case, LPC results may be interpreted as falsely positive. A potential reason for the different results in LPC could be the artificial lysosomal enlargement using vacuolin-1 treatment.

#### 2.7.3. The Average Maximum Potentiation Is Higher in Patch-Clamp Compared to SSME

Besides EC_50_, which is the most crucial value to identify potential drugs, the maximum potentiation of the target protein activity in the presence of the compound is very relevant for assessing the quality of a compound. Comparing across the compounds for which we could conclude a reliable potentiation from the EC_50_ fit (Figure 8, Table 4), we found that the average potentiation is somewhat larger in patch-clamp recordings compared to SSME. In whole-cell APC, LPC, and SSME, we observe average potentiation of 409 ± 309%, 276 ± 160%, and 97 ± 47%, respectively.

Despite the lower potentiation, the reliability and quality of the fits in SSME seem to be generally higher: the average relative error of the potentiation obtained from the fits is lowest for SSME data (14.6 ± 8.3%), followed by LPC data (18.7 ± 8.3%). The value obtained with APC data is significantly higher (82 ± 65%). The adjusted R² representing the quality of the fits is highest in SSME (0.978 ± 0.017) and significantly lower in both the APC (0.937 ± 0.054) and LPC (0.934 ± 0.072) datasets.

### 2.8. Assay Parameters May Affect Apparent Drug Potencies

To gain a better understanding of the effects of assay parameters on the apparent compound potency that may vary across different technologies—such as K^+^ concentrations, pH values, the presence of pH gradients, and the orientation of TMEM175 in the membrane—we conducted additional control experiments utilizing the SSME technology.

#### 2.8.1. The Driving Force Has No Effect on the Apparent Compound Potency

In our SSME assay, the K^+^ concentration gradient serves as the sole driving force. To investigate the potential impact of the applied K^+^ concentration on apparent drug potency, we employed K^+^ concentrations of 2 mM, 5 mM, 10 mM, and 50 mM to induce K^+^ flux through TMEM175 and examined the relative current potentiation in the presence of varying concentrations of DCPIB (Figure 9A). When lower K^+^ concentrations are employed, larger standard deviations are observed due to a lower amplitude of the currents (compare Figure 3). However, regardless of the K^+^ concentration used, the average potentiation remains consistent for each DCPIB concentration tested. The apparent EC_50_ for DCPIB is not affected by the K^+^ concentration, indicating that the determined EC_50_ values are independent of the driving force. These findings also show that current artifacts upon K^+^ concentration jumps that are significant at 50 mM concentration, but not at 5 mM concentration (compare Figure 3D), do not affect the results and are negligible.

**Figure 9 ijms-24-12788-f009:**
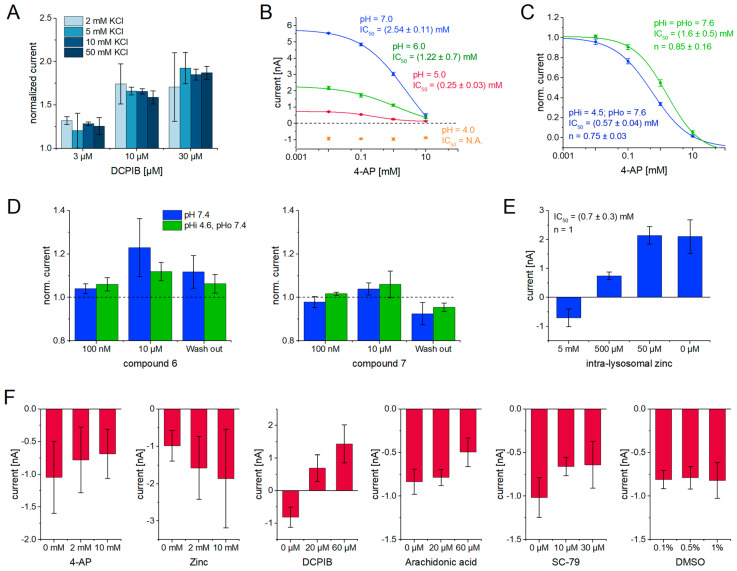
Effects of pH, pH gradients, and protein orientation on apparent compound potencies in SSME recordings and off-target compound effects: (**A**) Effects of 3 µM, 10 µM, and 30 µM DCPIB on TMEM175 currents activated using K^+^ concentration jumps employing different K^+^ concentrations. The non-activating solution contained 140 mM NaCl, the activating solution contained x = 2, 5, 10, or 50 mM KCl and (140 − x) mM NaCl. Currents were recorded using one 96-well sensor plate with the SURFE^2^R 96SE. Currents were normalized to the currents recorded with the respective K^+^ concentration in the absence of DCPIB. The bar plot shows average, normalized currents and SEM recorded from N = 8 sensors per condition; (**B**) pH dependence of IC_50_ curves for 4-AP recorded using TMEM175 lysosomes with the SURFE^2^R N1. A solution exchange of 50 mM Na^+^ against 50 mM K^+^ was used to stimulate TMEM175. Datasets for each pH were measured on different sensors. Datasets were normalized to the peak current recorded in the absence of 4-AP before averaging. To indicate the pH dependence of absolute currents, we normalized each pH dataset a second time, using the average current obtained in the absence of 4-AP (I_max_). Average peak currents and SEM, recorded from N = 5 sensors, are shown. I_max_ was fixed in the subsequent fitting with the equation I = I_max_ − (I_max_ − I_min_)/(1 + (c/EC_50_)^n^). The dataset recorded at pH 4.0 (orange) only consists of artifact currents that are independent of 4-AP concentration; (**C**) As in (**B**), but instead of pH values being equilibrated in both internal and external solutions, we applied a pH gradient as indicated, incubating with non-activating solution at pH 4.5 for 3 min to equilibrate the intra-lysosomal pH, before switching to non-activating solution at pH 7.6 defining the external pH ~700 ms in advance of the 50 mM K^+^ jump to stimulate TMEM175. Both datasets—symmetrical pH 7.6 (green) and pH gradient (blue) conditions—were recorded on the same sensor, with N = 5 sensors in total. Since no major impact of the pH conditions on I_max_ have been observed, averaged and normalized currents are shown (compare with Figure 4G); (**D**) Effect of pH gradients on compound potency recorded with the SURFE^2^R N1. The compounds are blinded, but the labels match with the labels shown in Figure 8. Experimental conditions as described in (**C**). Currents in the presence and the absence of a pH gradient were recorded with different sensors. Each sensor was used for four measurements in the following sequence: in the absence of compound, in the presence of 100 nM compound, in the presence of 10 µM compound, and in the absence of compound (wash-out). Datasets were normalized to the peak current recorded in the absence of compound before averaging. Average peak currents and SD from N = 3 sensors are shown; (**E**) Intra-lysosomal zinc application and its effects on TMEM175 recorded with the SURFE^2^R N1. Sensors were prepared with TMEM175 lysosomes that were pre-loaded via 1:10 dilution in zinc containing non-activating solution, followed by 10 pulses of sonication as described in the Methods section (Section 4.2.5). For each zinc concentration, we applied a solution exchange of 50 mM Na^+^ against 50 mM K^+^ in the absence of zinc, causing zinc to only be present inside the lysosomes. Average peak currents and SD from N = 5 sensors are shown, and no normalization was applied; (**F**) Measurements of compound effects on control lysosomes purified from HEK293 cells expressing endogenous TMEM175, recorded with the SURFE^2^R N1. A solution exchange of 50 mM Na^+^ against 50 mM K^+^ was used, both in the absence of the compound, and in the presence of the two highest compound concentrations used for EC_50_ and IC_50_ determination using the TMEM175 sample (Figure 6 and Figure 8A). The three different experimental conditions shown in one graph were applied on the same sensor. Average peak currents and SD from N = 4 sensors are shown, and no normalization was applied.

#### 2.8.2. Effects of pH and pH Gradients on Apparent Drug Potencies

Lysosomes exhibit a natural pH gradient across their membrane. We already found that a natural pH gradient compared to symmetrical pH 7.5 conditions does not significantly affect the K^+^ flux through TMEM175 (Figure 4D,G). However, it may affect other properties, potentially the efficacy of compounds, either in general, or for a subset of compounds that act via a distinct mechanism. To test whether the pH is a relevant parameter affecting the efficacy of compounds, we tested the effect of compounds on TMEM175 currents under different pH conditions using the SURFE^2^R N1 (Figure 9B–D).

We determined the IC_50_ for 4-AP at different pH values and found that the IC_50_ shifts from (2.54 ± 0.11) mM at pH 7.0, to over (1.22 ± 0.7) mM at pH 6.0, to (0.25 ± 0.03) mM at pH 5.0 (Figure 9B). At pH 4.0, no effect of 4-AP is observed, since there is no TMEM175 activity recorded at this pH (compare Figure 4C–D). We subsequently examined the impact of a pH gradient on the IC_50_. We determined IC_50_ values at pH 7.6 on both sides of the lysosomal membrane and in the presence of a natural pH gradient across the lysosomal membrane, with an external pH of 7.6 and an internal pH of 4.5 (Figure 9C). In the presence of a pH gradient, the IC_50_ slightly shifted to a higher potency, from (1.6 ± 0.5) mM to (0.57 ± 0.04) mM. Considering that the application of a pH gradient also generates a membrane potential over the lysosomal membrane due to H^+^ translocation through TMEM175, this small effect may be not a consequence of the pH gradient, but a consequence of a small change in voltage. It has been shown that 4-AP binding is voltage-dependent [28,45].

We further investigated the influence of pH gradients on the apparent potencies of two enhancers on the SURFE^2^R N1 (Figure 9D). These enhancers exhibited extremely high potency in LPC recordings, but did not demonstrate comparable potencies in whole-cell APC and SSME recordings (compounds **6** and **7**, Figure 8D). One major difference is the physiological pH gradient in LPC, while SSME and APC recordings were performed in the absence of a pH gradient. No significant effects of a pH gradient on the current potentiation were detected, indicating that the discrepancy in apparent compound potencies between technologies cannot be attributed to the presence of a pH gradient.

In summary, multiple lines of evidence indicate that the apparent potency of compounds is not significantly influenced by pH or pH gradients. The need for considering pH gradients in an HTS assay for characterizing pharmacological parameters of drug-TMEM175 interactions may be questioned.

#### 2.8.3. Effects of TMEM175 Orientation on Drug Potency

In our whole-cell APC assay, compounds are applied with the external solution. Thus, any immediate effects on TMEM175 are a consequence of compound interaction with the lysosomal side of TMEM175. In the SSME experiments, compounds are added from the external side as well, but since lysosomes are used, compound effects on TMEM175 activity originate from interactions with the cytosolic side of TMEM175. Discrepancies of compound effects across SSME and APC recordings may therefore be a consequence of different orientations of TMEM175 inside the target membrane.

To examine this, we conducted SSME experiments utilizing lysosomes that were pre-loaded with varying concentrations of zinc (Figure 9E). This enabled us to investigate the effects of zinc on TMEM175 activity when it interacts with the lysosomal side of TMEM175, closely simulating the conditions observed in whole-cell APC experiments (Figure 7E). Upon applying zinc from the lysosomal side, we observed inhibition with a comparable potency (IC_50_ = (0.7 ± 0.3) mM) to experiments where zinc was applied from the external side (IC_50_ = (1.53 ± 0.21) mM, Figure 6D). Similar to the external application, we did not observe any effect of zinc at a concentration of 50 µM, which is close to half-saturation of the TMEM175 enhancement found in the APC assay (Figure 7E). Our findings are in general agreement with literature findings that pore blockers have similar effects when applied with internal and external solutions in patch-clamp recordings [1] and exclude the possibility that TMEM175 orientation is the reason for the discrepancy of the found zinc effects on TMEM175 in whole-cell APC and SSME recordings.

### 2.9. Off Target Effects in SSME Recordings

Potential discrepancies of apparent drug potencies across technologies may be due to different overlaying off-target effects. The APC recordings on control HEK293 UT cells show minor off-target effects, which may be negligible for analysis (Figure 7F). In SSME recordings, the presence of high compound concentrations may alter the solution exchange artifacts—here, the capacitive currents are recorded when Na^+^ is replaced with K^+^, reflecting ion binding to the lipid membrane [41]. The alteration of artifact currents would consequently lead to false positive results or shifts in apparent EC_50_ or IC_50_ values when investigating compound effects on the TMEM175 sample. To identify potential off-target effects in SSME recordings on the TMEM175 sample, i.e., effects not related to TMEM175, we treated control lysosomes purified from the parental HEK293 cell line with DMSO, the tool compounds used in this study (4-AP, zinc, DCPIB, arachidonic acid, SC-79), and the eight blinded test compounds.

#### 2.9.1. SSME Recordings Tolerate DMSO Concentrations of 1%

In all SSME recordings conducted in the presence of compounds, we used 0.1% (*v*/*v*) of DMSO. To exclude potential interferences of DMSO with our recordings, we investigated the effect of 0.1%, 0.5%, and 1% (*v*/*v*) DMSO on the currents obtained from control lysosomes recorded with the SURFE^2^R N1 (Figure 9F, right). We found no effects on recorded artifact currents, independent of the DMSO concentration used.

#### 2.9.2. Off-Target Compound Effects for Tool Compounds Are Mostly Negligible

To explore the potential off-target effects of the tool compounds 4-AP, zinc, DCPIB, arachidonic acid, and SC-79, we applied the two highest compound concentrations used to determine their apparent potencies with lysosomes overexpressing TMEM175 (Figure 6 and Figure 8A) to control lysosomes (Figure 9F) using the SURFE^2^R N1. We found off-target effects that differ in amplitude and type of effect (potentiation or inhibition). However, in all cases, the effects are small compared to the effects on the TMEM175 sample and may have no or only a minor impact on the determined apparent drug potencies.

For example, the average artifact current recorded from a 50 mM K^+^ concentration jump changes in the presence of 10 mM 4-AP from (−1.06 ± 0.27) nA to (−0.69 ± 0.37) nA, indicating an off-target signal potentiation of +380 pA (Figure 9F). In contrast, TMEM175 currents decrease from (3.48 ± 0.19) nA to (−1.22 ± 0.1) nA (Figure 6E). The off-target potentiation effect is only ~8% compared to the effect on the TMEM175 lysosomes and in the opposite direction. In contrast to 4-AP, the application of zinc to control lysosomes leads to a small decrease in current, indicating off-target inhibition (Figure 9F).

Interestingly, SC-79 also reveals a small off-target effect, changing the control current from (−1.02 ± 0.23) nA to (−0.64 ± 0.27) nA in the presence of the 30 µM compound (Figure 9F). The absolute off-target current potentiation correlates well with the small potentiation effect observed in SSME recordings, allowing the conclusion that SC-79 has no direct effect on TMEM175 activity, potentially due to the lack of PKB in SSME recordings. The determined EC_50_ using TMEM175 lysosomes rather reflects the interaction of SC-79 with the lipid membrane, changing its capacitive properties (Figure 8A).

The highest off-target compound effect was observed for DCPIB. 20 µM DCPIB changes the control current from (−0.81 ± 0.31) nA to (+0.69 ± 0.41) nA (Figure 9F), corresponding to ~35% of the compound effect found with lysosomes overexpressing TMEM175 (Figure 6F). DCPIB is a small aromatic compound that inserts into biological membranes [49], potentially changing their structural integrity and electric properties. Previous studies have demonstrated the problematic nature of these substances, which can also generate artifacts in SSME recordings [34,50].

Note that for proper off-target effect consideration, data recorded using the same platform should be used. We also investigated the effect of zinc and DCPIB on control lysosomes expressing endogenous TMEM175 using the SURFE^2^R 96SE (Appendix A). For zinc, off-target effects recorded with both SURFE^2^R instruments are very similar, while for DCPIB, off-target inhibition was observed, potentially resulting in an underestimation of the potentiation found in SSME recordings. When correcting for these off-target effects, the potentiation of DCPIB in SSME recordings (275%, Figure 6F) is somewhat closer to the potentiation observed in APC recordings (680%, Figure 7E). Note that the EC_50_ determined for DCPIB using the SURFE^2^R 96SE (9.6 µM, Figure 6F) is not significantly affected by these off-target effects, since it matches well with the EC_50_ concluded from APC (16.7 µM, Figure 7E).

#### 2.9.3. A Set of Test Compounds Shows Significant Off-Target Compound Effects

We also investigated the effect of the eight test compounds on the artifact currents recorded with control lysosomes using the SURFE^2^R 96SE. The four highest compound concentrations used to generate the dose–response curves shown in Figure 8 were used.

Artifact currents are typically in the range of −1 nA, while TMEM175 sample currents are +4 nA (Figure 1K). The presence of a compound typically leads to a change in the artifact’s current magnitude by 100 pA to 1 nA. This corresponds to 3% to 30% of the current recorded with the TMEM175 sample in the absence of the compound. For each compound concentration, the corresponding percentage is indicated in the graph to show the fraction of the false or off-target compound effect on the current observed with the TMEM175 sample (Figure 8). Some compounds reduced the artifact current obtained with control lysosomes, essentially representing off-target inhibition, but most compounds generated a positive amplitude, which led to an increase in the artifact current, representing off-target potentiation.

Out of the eight compounds tested, four demonstrate off-target effects below 10% at the highest concentration of 100 µM. These effects can be disregarded for further analysis. The remaining four compounds exhibit off-target effects ranging from 20% to 40% at their highest concentrations. In this case, the off-target effects should be taken into consideration when analyzing and interpreting the modulation of TMEM175 at high concentrations of the test compound.

As an example, compound **3** (cluster 1) elicits a potentiation of 74% in the TMEM175 sample. However, there is an off-target potentiation of 40% observed in the control sample. This off-target effect becomes noticeable between 30 µM and 100 µM, potentially leading to a right-shift in the apparent EC_50_. Notably, the EC_50_ obtained from the SSME assay is 30 µM, which is higher than the EC_50_ values obtained from the APC (7 µM) and LPC (12 µM) assays. Nevertheless, it is important to note that even though compounds with significant off-target effects in the SSME assay can impact the overall response, they are not responsible for the observed discrepancies in EC_50_ values or potentiation observed with different technologies. In addition, the presence of off-target effects in the SSME assay should not be considered solely as artifacts, but should be regarded as providing valuable additional information. Compounds that induce off-target effects by directly interacting with biological membranes, thereby impacting their structural integrity and altering overall electrical properties, are unsuitable for human treatment at the given concentration. Hence, further comparative analysis with more potent and efficacious modulators will be crucial.

## 3. Discussion

Here, we present a functional characterization of TMEM175 in lysosomes using SSM-based electrophysiology, revealing the kinetic properties of K^+^ and H^+^ translocation and the effect of pH on TMEM175 conductivity. Based on this knowledge, we developed an HTS-compatible assay mode to study and characterize compounds that modulate TMEM175 using SSME. We also compared the results obtained from this assay with those obtained from whole-cell APC and LPC recordings, providing a comprehensive perspective on the compound effects on TMEM175 across different experimental approaches and assay conditions.

### 3.1. Lysosomal TMEM175 Conducts Both H^+^ and K^+^ at Similar Rates

Several authors independently found that TMEM175 conducts K^+^. Most studies applied the whole-cell patch-clamp technique [2,28,45], but similar results were obtained from radioactive uptake assays using proteoliposomes [3] or from LPC [1]. The physiological role, however, remained speculative. It was suggested that K^+^ conductivity is required to regulate the lysosomal membrane potential that may be required to regulate the fusion of lysosomes with other organelles [1]. Recently, two studies suggested that TMEM175 is a proton channel [6,42]. Hu et al. could not reproducibly detect K^+^ currents [6]. They described the physiological role of TMEM175 as a pH regulator.

To examine the selectivity of TMEM175, we applied concentration jumps of different cations in an SSME assay and found that TMEM175 conducts K^+^, Rb^+^, and Cs^+^, but not Li^+^, Na^+^, or choline^+^ (Figure 2). We found that the Cs^+^ flux through TMEM175 was faster compared to the K^+^ flux (Figure 2D,F). This observation aligns with previous findings of a two-fold difference between K^+^ and Cs^+^ permeabilities [1] and currents [28]. From the relative currents recorded with K^+^ and Na^+^, we estimate that the K^+^ translocation rate is at least 200-fold faster than the Na^+^ translocation rate, which is below the detection limit of the technique (Figure 2D,F). Other authors found a ratio of 9 [4] and 29 [28] using whole-cell patch-clamp, and a ratio of 36 using endolysosomal patch-clamp [1]. In the SSME assay on lysosomes, the apparent selectivity for K^+^ over Na^+^ is further increased.

In addition, we found that protons are translocated through TMEM175 (Figure 5). We determined the relative permeabilities for K^+^ and H^+^ and found ratios of P_H_/P_K_ = 74,000 and P_H_/P_K_ = 23,000, depending on the K^+^ concentration range (Figure 3E), which fits with previously reported results [6,42]. In fact, the average P_H_/P_K_ ratio is 48,500 and essentially identical to the one found by Hu et al. (48,000), validating the comparability of the results obtained with SSME and those found in previous patch-clamp studies.

The permeability ratios suggest that H^+^ is the dominant species that is conducted by TMEM175. However, to conclude which ion is predominantly translocated at physiological conditions, the natural driving forces need to be considered [51]: the lysosomal pH is well regulated at pH 4.6, while the cytosolic pH is 7.2, corresponding to a Δ[H^+^] = 31.6 µM across the lysosomal membrane. On the other hand, the lysosomal K^+^ concentration is 2–50 mM, while the cytosolic K^+^ concentration is 140 mM, corresponding to a chemical gradient of Δ[K^+^] = 90–138 mM across the lysosomal membrane. The corresponding ratio of the concentration gradients across the lysosomal membrane is Δc_K_/Δc_H_ = 2850–4350. Placing the natural driving forces into context with the relative permeabilities (P_H_/P_K_ = 23,000–74,000), we can determine relative fluxes J_H_/J_K_ mediated through TMEM175. Depending on the exact conditions, the efflux of a single H^+^ occurs between 5 to 26 times more frequently than the influx of a single K^+^. Considering the lysosomal membrane potential between −20 mV and −40 mV, K^+^ influx would be in favor over H^+^ efflux, leading to very similar H^+^ and K^+^ translocation rates through TMEM175. Our findings therefore support that—considering physiological driving forces—TMEM175 conducts both K^+^ and H^+^ at similar rates and that both fluxes may have their own physiological relevance, as proposed by previous authors.

### 3.2. TMEM175 Conductivity Is a Function of Cytosolic pH and K^+^ Concentrations

While literature findings suggest that the ion permeability of TMEM175 is affected by lysosomal pH [6], our SSME recordings uncovered the effects of cytosolic H^+^ and K^+^ concentrations on the ion permeabilities of TMEM175.

The I/c curve for K^+^ revealed two different conductivity states (Figure 3E). When low K^+^ concentrations (<20 mM) are applied with the external solution, the I/c relationship reveals a four-fold increased permeability compared with the I/c curve recorded for K^+^ concentrations > 100 mM. Since the cytosolic K^+^ concentration is well regulated at 140 mM, K^+^ translocation through TMEM175 naturally occurs with the lower permeability. However, as a response to hypokalemia and an increased K^+^ release into the intracellular space [52], the permeability of TMEM175 for K^+^ may increase to keep up lysosomal K^+^ uptake, suggesting a physiological relevance of TMEM175-mediated K^+^ influx into the lysosomes.

We also found a large impact of cytosolic pH on both K^+^ and H^+^ permeability through TMEM175. First, the pH dependence of K^+^ flux through TMEM175 reveals a bell-shaped activity curve with two pK values for acidic downregulation (Figure 4D). The TMEM175 translocation rate drops between pH 8 and pH 6 to only 30% of its maximum rate. When altering only the lysosomal pH, the activity of TMEM175 remains largely unaffected within this pH range (Figure 4E). Consequently, it can be concluded that the acidic downregulation observed with pK = 7 is primarily a consequence of changes in cytosolic pH.

Similar effects have been found recently and could be attributed to pH-dependent structural changes within the K^+^ permeation pathway [42]: at pH 7.4, the TMEM175 structure reveals three ordered K^+^ binding sites, but only one weak K^+^ site is observed at pH 5.5; this finding is surprisingly consistent with our experimental observation that K^+^ flux at pH 6.0 is reduced to 30% compared to pH 8.0. Zheng et al. stated that the reduced number of K^+^ binding sites at acidic pH may be a result of competition with H^+^ fluxes. However, pure competition between K^+^ and H^+^ fluxes through TMEM175 is unlikely to be the sole cause for the observed acidic downregulation of K^+^ flux, because two pK values are required to describe this process (Figure 4D). We propose the downregulation of K^+^ flux with pKa = 4.5 is a result of competition between K^+^ and H^+^ fluxes. At elevated H^+^ concentrations and in the absence of a driving force for net H^+^ translocation (0 mV, ΔpH = 0), TMEM175 still facilitates bidirectional—hence, electroneutral and indetectable—steady-state H^+^ flux. This H^+^ flux effectively impedes the directional K^+^ flux, which is recorded. Comparing permeability ratios and ion concentrations, it becomes apparent that the competition between H^+^ and K^+^ fluxes only become significant at this particular pH (Section 3.1). Conversely, the downregulation of K^+^ flux with pKa = 7.0 may be attributed to H^+^ binding to the cytosolic site of TMEM175 through an unknown mechanism, reducing its ability to conduct ions.

Interestingly, cytosolic pH also effects the I/c curves for H^+^ flux: when performing H^+^ concentration jumps, we only find a linear I/c relationship when the external pH is close to the physiological pH range (Figure 5E,J), indicating that the pH-dependent structural changes also affect H^+^ permeation. Cytosolic acidification sets boundaries for the H^+^ conductivity, independent of the direction of H^+^ translocation.

The major effect of cytosolic pH on TMEM175 conductivity is within the physiological pH range, indicating a physiological relevance of the observed pH dependence. Cytosolic acidification reduces the ion conductivity of TMEM175, preventing further H^+^ efflux from the lysosomes, which prevents further acidification of the cytosol. TMEM175 could play a role in cytosolic pH regulation by fine-tuning the release of H^+^ from lysosomal H^+^ storage.

### 3.3. Lysosomal Acidification Does Not Stimulate TMEM175 Activity

In contrast to our findings that the cytosolic pH controls both the K^+^ and H^+^ permeability of TMEM175 (Section 3.2), Hu et al. demonstrated that lysosomal acidification is a prerequisite for the H^+^ permeability [6], concluding a strict gating mechanism. From our SSME recordings, we were unable to observe an impact of lysosomal pH that could explain these conclusions. From our H^+^ flux assays, we concluded similar H^+^ permeabilities for both H^+^ influx and efflux, and the H^+^ permeability is unaffected when lysosomal pH drops from pH 7.6 to pH 4.6 (Figure 5E,J). In contrast, we do find a linear increase in the K^+^ flux through TMEM175 of about ~20% when the intra-lysosomal pH is acidified from pH 8 to pH 4.5 (Figure 4G). However, this effect is not as dramatic as those found by Hu et al. and is likely a consequence of the H^+^ flux following the application of the pH gradient, which generates a small voltage across the lysosomal membrane, acting as an additional driving force for the K^+^ flux, which is recorded.

Regardless of this major discrepancy, several findings reported by Hu et al. were successfully replicated using SSME, i.e., the P_H_/P_K_ ratio (Section 3.1) and the inhibition of K^+^ flux at acidic pH (Section 3.2). Our findings also align with the discovery by Hu et al. that TMEM175 undergoes downregulation when the natural lysosomal pH gradient is reversed. However, we attribute this downregulation to effects of decreasing cytosolic pH rather than increasing lysosomal pH. Other findings by Hu et al. cannot be experimentally replicated using SSME, for example, the increased H^+^ conductivity when both sides of the membrane are equally acidified. For this experiment, voltage is required as a stimulus and SSME is restricted to concentration gradients.

To summarize, analyzing the kinetic properties of TMEM175 in terms of H^+^ and K^+^ translocation and their pH dependence is challenging due to the presence of multiple interdependencies: pH adjustments will not only affect the potential H^+^ regulation of TMEM175, but also induce H^+^ flux. Investigating the pH dependence of K^+^ fluxes will always be biased due to competition with H^+^ fluxes. Finally, protons have different effects on TMEM175 depending on lysosomal or cytosolic application, and experimental limitations such as the stability of pH gradients and effects of pH on other membrane components may lead to additional obstacles. Further investigations are required to fully understand the impact of pH on H^+^ and K^+^ fluxes through TMEM175.

### 3.4. TMEM175 Tool Compounds Reveal Different Apparent Potencies in HTS Compatible APC and SSME Assays

TMEM175 is located in a highly significant risk loci for PD on chromosome 4 [53] and it was shown recently that TMEM175 deficiency results in unstable lysosomal pH and α-synuclein aggregation, leading to PD pathogenesis [5]. Thus, TMEM175 represents a putatively important drug target for PD modification.

Two high-throughput screening (HTS) assays have been developed to enhance TMEM175 drug development and screening, employing SSME (SURFE^2^R 96SE) and whole-cell APC (SyncroPatch 384i), respectively. Both technologies have distinct capabilities. The combination enabled us to investigate drug effects on TMEM175 located in both plasma membrane and lysosomes using either voltage or concentration gradients to stimulate TMEM175 activity. We found that compound effects were different across both techniques, which is elaborated in the following sections.

#### 3.4.1. 4-AP

In APC, we determined an IC_50_ value for 4-AP of (17 ± 8) µM (Figure 7E), which is slightly lower compared to values found in the literature, ranging from (19 ± 7) µM for Cs^+^ flux [45], (35 ± 15) µM for K^+^ flux [1], (55 ± 13) µM for H^+^ flux [45], to (79 ± 1) µM for K^+^ flux [28]. IC_50_ values may also depend on the ion species that permeates through the channel. Considering that our IC_50_ value reflects the potency of 4-AP to inhibit Cs^+^ flux (Figure 7A), the comparable literature IC_50_ value is very close to our value (17 and 19 µM, respectively).

In contrast, the IC_50_ values determined with SSME ((1.67 ± 0.3) mM) were 100-fold higher compared to the value obtained with APC, and 20 times higher compared to the largest literature value observed for K^+^ flux (which is the condition used for SSME recordings), indicating a lower potency of 4-AP in SSME (Figure 6E). Interestingly, we found that acidification (Figure 9B) and natural lysosomal pH gradients (Figure 9C) slightly increase the potency of 4-AP, but the effect of pH is not sufficient to explain the observed discrepancy of apparent compound potency between the APC and SSME recordings. However, it has been shown previously that the interaction between 4-AP and TMEM175 is voltage-dependent [28,45], and voltage control is a key difference between APC and SSME recordings.

#### 3.4.2. Zinc

In our whole-cell APC recordings, we observed that the presence of zinc significantly increases TMEM175 activity by an average of 301 ± 94% when administered at a concentration of 0.5 mM. We found an EC_50_ of (102 ± 12) µM (Figure 7E). This is in contrast to findings from the literature, which have shown that zinc blocks TMEM175 with a mechanism similar to that of 4-AP [2]. However, the found EC_50_ value in our APC recordings is similar compared to the IC_50_ value found in the literature using whole-organelle patch-clamp, which is (38 ± 13) µM [1].

In our SSME recordings, we confirmed the inhibitory effect of zinc (Figure 6D). However, similar to 4-AP, we found an IC_50_ value of (1.53 ± 0.2) mM, which is 50-fold higher compared to the literature value. Interestingly, both pore blockers, 4-AP and Zn, show similar IC_50_ values in literature [1] and also in SSME recordings. This implies that the larger IC_50_ values in SSME are not compound-specific (potentially indicating a false result), but are specific for the pore-block mechanism. The missing voltage in SSME recordings could be one possible explanation.

To exclude the potential inhibitory effects of zinc that may be revealed in APC recordings using higher zinc concentrations, we also tested up to 10 mM zinc in APC, which was shown to generate a full block in SSME recordings on lysosomes. We could not find any changes in the currents when the zinc concentrations increased from 0.5 mM to 10 mM in APC.

It has been found before that the block by zinc and 4-AP [1] occurs regardless of extracellular or intracellular application [3], which we also confirmed via SSME (Figure 9E). The different effect of zinc on TMEM175 activity in APC and SSME therefore is not a consequence of a different TMEM175 orientation in the plasma membrane compared to lysosomal membranes.

Note that the inhibitory effect of zinc found in the literature and SSME recordings is based on experiments with lysosomes, while the enhancing effect found in whole-cell APC is based on recordings of TMEM175 located in the plasma membrane. However, it seems unlikely that the plasma membrane itself causes these discrepancies. First, we know from whole-cell recordings of several TMEM175-overexpressing cell lines that certain lines exhibited current blockade while others demonstrated current enhancement upon the application of zinc. Second, 4-AP interacts with TMEM175 through a very similar mechanism as zinc [2], but still blocks TMEM175 in our APC assay. We also ruled out the off-target effects of zinc application in both APC (Figure 7F) and SSME recordings (Figure 9F). We hypothesize that zinc may activate TMEM175 in whole-cell APC recordings of some cell lines due to its effects on the AKT pathway [54], similar to SC-79. This is supported by the fact that the PKB-mediated effects of SC-79 on TMEM175 were not consistently reproduced across different cell lines as well [6].

Further experiments are needed to fully clarify the origin of zinc enhancement in whole-cell APC recordings.

#### 3.4.3. DCPIB

Hu et al. found that 50 µM of DCPIB leads to an eight-fold increase in K^+^ flux through TMEM175 at +80 mV [6]. In our APC recordings, we found a 6.8-fold potentiation with a half saturation of EC_50_ = (16.7 ± 11.3) µM (Figure 7E). With SSME, we found a reduced potentiation compared to APC, which is 2.8-fold (Figure 6F). The underestimation may be partly a consequence of off-target inhibition found with control lysosomes (Appendix A). However, the determined EC_50_ of (9.6 ± 5.1) µM matches with the EC_50_ obtained using whole-cell APC. Overall, the available data on DCPIB potency does match very well across different techniques and experimental conditions.

#### 3.4.4. Arachidonic Acid

Hu et al. found that 100 µM of arachidonic acid leads to a 14-fold increase in K^+^ flux through TMEM175 in whole-cell configuration at +80 mV [6]. In our APC recordings, we found an EC_50_ of (83 ± 20) µM and a 34% increase in TMEM175 activity (Figure 7E), which is almost two orders of magnitude below the literature reference. Here, we utilized 0.05% BSA to stabilize the current read-out, which proves advantageous for extended compound application protocols. BSA is known to bind arachidonic acid [47]. However, APC recordings in the absence of BSA did not affect the recorded EC_50_ or potentiation. The origin of the discrepancy between our APC results and the literature findings remains elusive.

SSME recordings reveal a 2.3-fold potentiation of the TMEM175 current (Figure 6G), which is higher compared to our APC recordings, but lower compared to the literature findings. Using SSME, we observed a 40-fold higher apparent affinity (EC_50_ = (2.3 ± 0.5) µM) compared to APC. Since off-target effects in SSME could be excluded (Figure 9F), the difference between our APC and SSME recordings may be due to different experimental conditions, most prominently the voltage application or membrane environment.

#### 3.4.5. SC-79

SC-79 is a compound that was described to enhance TMEM175 activity. It binds to protein kinase B (PKB) to stabilize the open conformation of the kinase, which, in turn, induces channel opening via conformational coupling [28,48]. Wie et al. found that 10 µM of SC-79 led to a two-fold increase in TMEM175-mediated K^+^ flux at +80 mV [48]. In our APC recordings, 50 µM SC-79 enhanced the TMEM175 current to 305 ± 39% at +100 mV (Figure 7F) and from the dose dependence, we found an EC_50_ of (62 ± 42) µM and a maximum potentiation of 345 ± 102% (Figure 8A).

In SSME recordings, the effect of SC-79 was only 9 ± 4% (Figure 8A), which could be fully attributed to off-target effects using measurements on control lysosomes (Figure 9F). Given that SSME recordings employ enriched lysosomes as samples, stored at −80 °C, we did not expect an effect of SC-79 on TMEM175 activity using this in vitro technique. PKB is not present in the assay. However, it was described that PKB forms a stable complex with TMEM175 [48], which may partially endure lysosomal purification. Our results indicate that this complex did not survive lysosomal purification according to Schulz et al. [38] and storage at −80 °C. However, the mechanical procedure to isolate lysosomes for LPC is sufficient to keep the complex and detect the effect of SC-79 (Figure 8A), which was similar to that observed in whole-cell APC. The chemical treatment using vacuolin-1 to increase the lysosomal size did not disrupt the interaction between TMEM175 and PKB. It seems that, compared to the lysosomal preparation for SSME, the lysosomal preparation for LPC is a gentler method that allows proteins to remain attached to the organelle, while being more labor-intensive as it needs to be performed freshly before each LPC recording.

### 3.5. Eight Blinded Test Compounds Generated Different Results in APC, SSME, and LPC

While APC and SSME recordings using the tool compounds 4-AP, zinc, DCPIB, arachidonic acid, and SC-79 partially produced different half-maximal concentrations and values for maximum potentiation, we could not find systematic differences across both techniques, i.e., that one of the techniques reveals higher apparent affinities or potentiation over the other (Table 3). To statistically compare compound effects across technologies, we increased the number of compounds tested and also added LPC as a third technology (Figure 8): for a set of eight blinded test compounds, we compared apparent compound potencies across SSME recordings using purified lysosomes which are partly contaminated with plasma membrane vesicles (Figure 1B), whole-cell APC recordings on TMEM175 located in the plasma membrane, and manual LPC.

#### 3.5.1. Maximum Potentiation Is Higher in APC and LPC Compared to SSME

Comparing different compounds across different assay technologies, we found evidence that the current potentiation is generally smaller in SSME recordings compared to patch-clamp. Which potentiation effect is closer to reality still needs to be questioned. In patch-clamp experiments, the observed higher potentiation factor may be attributed to the higher voltage utilized, which does not accurately reflect the in vivo situation where membrane potentials across the lysosomal membrane typically range between −20 and −40 mV. In contrast, our findings using SSME demonstrate that the extent of the driving force does not influence the potentiation, as evidenced by testing the effect of DCPIB with varying K^+^ concentrations (Figure 9A), which contradicts this hypothesis. The differences in potentiation could also be a consequence of the different read-outs, which is the steady-state current in patch-clamp and capacitive currents in SSME recordings.

It is also important to recognize that the potentiation observed in the human body may not always correspond directly to the potentiation measured under specific assay conditions using any assay technology. The human body is a complex system with various factors that can influence compound potency, including metabolic processes, tissue distribution, and interactions with other molecules. These additional effects cannot be fully assessed in vitro.

Furthermore, when evaluating the capabilities and assay windows of a particular assay technology, the average potentiation holds equal importance to the accuracy of the measurement. Our findings indicate that SSME surpasses APC and LPC in terms of the quality of EC_50_ fit, the accuracy of the determined maximum potentiation, the standard deviation of averaged data points, and z’ prime values (compare Figure 6 and Figure 7). This indicates a more accurate estimation of potentiation in SSME, demonstrating that the lower potentiation in SSME does not hinder the ability for hit identification or to rank compounds based on their potentiation.

#### 3.5.2. Apparent EC_50_ Values Differ across Assay Technologies

Similar to the tool compounds (Figure 6 and Figure 7), we found that the EC_50_ values for the eight blinded test compounds differ across methods, but we could not find systematic trends (Figure 8). This indicates that none of the three aforementioned methodologies are generally biased in one or another direction. However, we could group the eight test compounds into three clusters based on their EC_50_ values. Compounds in cluster 1 show similar apparent affinities, independent of the assay technology, while compounds in cluster 2 and cluster 3 show lower and higher apparent affinities with LPC compared to SSME and APC.

We attempted to identify the underlying reasons for the observed discrepancies (Figure 9). A notable distinction between the three assay technologies is that LPC involved the investigation of H^+^ fluxes in the presence of a pH gradient, whereas APC and SSME focused on examining Cs^+^ and K^+^ fluxes in the absence of pH gradients, respectively. For two compounds that exhibited a very high potentiation in LPC, but not APC and SSME (Figure 8, cluster 3), we performed SSME recordings in the presence of a pH gradient and found that the pH gradient has no impact on the apparent potency (Figure 9D). Additionally, for DCPIB, the apparent potentiation was very similar in H^+^ and Cs^+^ flux APC assays (Figure 7G). Since the IC_50_ of 4-AP is equally unaffected by pH and pH gradients (Figure 9B,C), we can conclude that pH and pH gradients do not significantly influence apparent compound potencies in our hands. As a conclusion, these factors might not need to be taken into consideration when establishing a screening assay.

Some parameters that differ across techniques cannot be experimentally addressed: one factor may be the presence of voltage in patch-clamp recordings, but not SSME. However, using SSME recordings and different K^+^ concentrations to stimulate TMEM175, we showed that the apparent potency of DCPIB is unaffected by the driving force (Figure 9A). Another important factor is the environment of TMEM175: the lipid and protein environment in APC recordings is defined by the plasma membrane composition; in LPC, it is defined by the lysosomal membrane. In SSME recordings, lysosomes are used, but our sample preparation still shows some impurities with plasma membrane vesicles (Figure 1B). Interestingly, the EC_50_ values determined with SSME are often between those determined with LPC and APC, indicating that the observed differences could indeed be a result of different lipid and/or protein environments within the samples.

To conclude, it remains uncertain whether the LPC results of cluster 2 and 3 compounds (Figure 8C,D) accurately reflect the true responses, particularly in cases where both APC and SSME recordings exhibit similar and distinct responses compared to the results obtained with LPC. This uncertainty arises because LPC may have its own biases, such as artificial membrane alterations caused by vacuolin-1 treatment, which could affect the observed outcomes. In contrast, the ability of SSME to identify false positive compound hits from other techniques was highlighted recently in the context of drug development on an amino acid transporter [36].

Undoubtedly, each technique has its own set of advantages and disadvantages. In the subsequent section, we will examine these factors and explore whether the pharmacological parameters derived from any of the techniques represent the true values.

### 3.6. LPC, APC, and SSME Are Complementary Technologies

For many compounds, tool compounds (Figure 6 and Figure 7), and blinded test compounds (Figure 8), we found different compound effects depending on the assay technology. The question remains as to which technique comes closest to reality and natural conditions, hence generating the most impactful results. In addition, compromises may be made to achieve higher throughput, quality of data, or assay flexibility. In the following paragraphs, the advantages and disadvantages of the three electrophysiological methods—whole-cell APC, manual LPC, and SSME recordings—are compared, which are also summarized in Table 5. Lysosomal APC development is underway, but it remains a work in progress. As a result, these three electrophysiological techniques are presently the preferred methods for characterizing the pharmacological properties of TMEM175.

#### 3.6.1. The Natural Driving Forces of TMEM175 Are pH and K^+^ Concentration Gradients

The natural driving forces over the lysosomal membrane are mainly defined by chemical gradients, i.e., an inward-directed K^+^ gradient of ΔK^+^ = 90–135 mM and an outward-directed H^+^ gradient of ΔpH = 2–3 units [51]. The membrane voltage is rather low and kept constant between −20 and −40 mV [51]. With SSME, ion flux through TMEM175 is stimulated at 0 mV using an adjustable pH (Figure 5) or K^+^ concentration gradient (Figure 3) in both directions, which can be set close to native conditions. In addition, the effects of a pH gradient on K^+^ conductivity (Figure 4) and vice versa may be analyzed. In contrast, patch-clamp studies employ voltage steps as a stimulus to drive ion translocation through TMEM175, and currents are measured at highly positive or negative voltages, even in LPC experiments, which exceed the native membrane voltages observed across lysosomal membranes. Overall, the measurement solution compositions in patch-clamp recordings are more restricted compared to SSME, since live cells allow for a smaller window of buffer variations compared to in vitro SSME recordings on lysosomes or plasma membrane vesicles.

#### 3.6.2. The Target Membrane and Protein Orientation May Affect Pharmacological Properties of TMEM175

In the present study, we compared different assay technologies operating with different samples. In whole-cell APC, one single cell is used to measure TMEM175 within the plasma membrane. In LPC, one single, enlarged lysosome is used and TMEM175 activity is measured in the native lysosomal environment. SSME is carried out using macroscopic samples, mostly containing lysosomes, with some degree of contamination with plasma membrane vesicles (Figure 1B). Interestingly, we found that the apparent potencies obtained with SSME are mostly between those obtained with LPC and APC (Figure 8), indicating the importance of the membrane environment.

Proteins within or attached to the target membrane can affect the functionality of a target protein [56]. The lipid environment within a membrane frequently exerts significant effects on membrane protein functionality, and it can also impact drug efficacy [57]. It was shown that the ion selectivity of TMEM175 differs between plasma membrane and endolysosomal patch-clamp [45]. But most importantly, a recent publication found a direct interaction between lysosomal LAMP proteins and TMEM175, very likely affecting potential protein/drug interaction interfaces and consequently drug potencies [29]. These observations explain the need for assays applicable for different target membranes and support the ongoing development of lysosomal APC, which is still in development. Manual LPC is an alternative when ultra-low throughput is sufficient, but the required treatment of lysosomes using vacuolin-1 to enlarge the vesicle size may have unwanted effects on the functional properties of the target protein. Vacuolin-1 was shown to interact with a certain protein to affect lysosomal trafficking [58]. However, direct effects on other proteins or biological membranes in general cannot be excluded. More research is required to estimate the potential effects of vacuolin-1 on the functional properties of lysosomal membrane proteins.

SSME recordings employ non-treated lysosomes of natural size resembling the most native chemical environment for the target protein. Macroscopic currents from millions of lysosomes are measured to increase signal-to-noise. However, purified lysosomes are not perfectly pure and contamination with plasma membrane vesicles was present (Figure 1B). Further optimization of the purification workflows will increase the purification efficiency, enabling researchers to compare the pharmacology of a target protein residing in different target membranes.

Besides the different chemical environments of TMEM175 within the respective target membranes, the orientation of TMEM175 in the plasma and lysosomal membranes is different. In APC, the luminal (i.e., intra-lysosomal) side of TMEM175 faces the extracellular environment. The application of drugs with the external solution in APC will therefore first affect the intra-lysosomal side of TMEM175. It is important to note that this side is distant from the initial accessibility of a drug upon human consumption.

Another challenge when measuring TMEM175 in whole-cell APC is the difficulty in acquiring native chemical gradients: the target cell needs to survive in pH 4.5 to enable measurements of TMEM175 under physiological conditions. These extreme pH conditions may also affect the physicochemical properties of lipids and proteins residing in the plasma membrane [59], with potentially unknown effects on target proteins.

#### 3.6.3. SSME and Patch-Clamp Recordings Employ Different Read-Outs with Impact on Data Interpretation

Since the approaches of patch-clamp and SSME are fundamentally different, the proper interpretation of the recorded data needs to be considered when results are compared across both techniques. Only in patch-clamp steady-state currents are obtained, while in SSME, capacitive currents are determined and peak currents reflect the initial velocity of the transport process, which are analyzed. Our data suggest that the different read-outs in patch-clamp and SSME do not affect kinetic parameters per se, as we do not see systematic differences, i.e., overall higher or lower EC_50_ values (Figure 8). However, in our study, we observed an overall lower current potentiation in SSME compared to patch-clamp recordings (Figure 8). This discrepancy may be attributed to the distinct read-out in SSME recordings, as other potential factors like variations in driving forces (Figure 9A) or differences in the target membrane—the average potentiation is similar in LPC and whole-cell APC recordings—are unlikely to be the cause.

Related to data interpretation, it is important to note that pre-steady-state currents have different origins in APC and SSME recordings. In SSME assays, pre-steady-state currents resulting from ion/substrate binding to the target protein may affect the peak currents, leading to shifted EC_50_ values, as shown recently for SGLT1 [40]. Here, peak currents reveal kinetic information about ion binding, enabling the conclusion of K_D_ values. In such cases, peak integrals can be considered as a preferable read-out for investigating transport and can be used for analysis instead. In contrast, pre-steady-state currents observed in APC recordings are a consequence of voltage steps and have a different molecular origin. The different molecular origins of pre-steady-state currents recorded with APC and SSME can be used as an important tool to understand the mechanisms of the target protein and potentially also the mechanism of compound/protein interactions.

However, we have no evidence for TMEM175—being a leak channel—to exhibit ion binding-associated pre-steady-state currents in SSME. Additionally, our APC recordings show no occurrence of pre-steady-state currents triggered by voltage steps, suggesting that this aspect is only relevant for other target proteins, such as membrane transporters or ligand-gated channels.

#### 3.6.4. Challenges of Assay Development in APC and SSME

Control measurements and assay development in APC and SSME recordings are similarly important, but with a different focus due to different read-outs. Understanding the challenges of each technique will help to achieve similarly high data quality for both APC and SSME recordings.

In patch-clamp, the steady-state baseline current before adding the compound is defined by the sum of all leak currents, which originate from the target protein, other proteins within the target membrane, and passive ion fluxes through the membrane. This is a consequence of the voltage being the major driving force, which acts on all ions present in internal and external solutions. In the case of a leak channel like TMEM175, the ionic composition of internal and external solutions needs to be adjusted to either match the specificity of the target protein and reduce off-target currents, or to generate a flux of a specific ion species that is to be investigated, i.e., H^+^ or K^+^. When off-target specificities overlap with that of the target protein, it may be challenging to find proper solution compositions. In addition, the efficient overexpression of the target protein may be required to increase the ratio between the target and off-target currents. Control measurements without the target protein are important to estimate the fraction of the recorded leak current mediated by the target protein.

In SSME experiments, leak currents are not present due to the capacitive read-out. The baseline current recorded prior to activation of the target protein using an ion/substrate concentration jump is zero. Unlike APC, the recorded current in SSME can be directly linked to the flow of a specific ion species, as the driving force applied is a concentration gradient of that particular ion, and voltage is not involved. Typically, adjusting the ion species in the activating solution suffices when investigating the flux of another ion, rendering assay modifications straightforward.

Despite the absence of leak currents in SSME, it is still possible for off-target effects to occur. Activation through concentration jumps can generate artifact currents primarily due to ion–membrane interactions [41], leading to potential over- or underestimation of the current associated with the target protein. It is crucial to include control measurements without the target protein to identify and quantify these artifact currents. In SSME, assay development focuses on minimizing these artifact currents [33]. If present, these artifact currents can be subtracted from the currents recorded with the target protein, as demonstrated in this study (Figure 2, Figure 3, Figure 4 and Figure 5; Appendix A).

During the screening of compounds using both APC and SSME, it is important to consider the potential influence of compound–membrane interactions or interactions between compounds and off-target proteins on the recorded currents. Additional measurements may be necessary to assess the effects of compounds on controls without the target protein. Notably, our results indicate that relative off-target compound effects are more pronounced in SSME recordings (Figure 8 and Figure 9F) compared to APC recordings (Figure 7F), emphasizing the increased significance of control measurements in SSME recordings. However, these effects typically start at high compound concentrations (>20 µM), likely not relevant for human treatment. Additionally, off-target compound effects in SSME recordings imply major changes to the structural integrity of biological membranes, indicating that the compound is not suitable for human treatment at the given concentration.

#### 3.6.5. SSME Provides Superior Assay Flexibility

In terms of assay flexibility, SSME offers several advantages over APC, primarily due to the limitation of APC to live cells. In SSME, internal and external measurement solutions can be varied beyond native limitations, allowing a larger window for assay optimization, achieving higher signal-to-noise and target-to-off-target effect ratios. While the application of large pH gradients may be problematic in whole-cell patch-clamp, SSME allows the application of buffers with pH values ranging from pH 3 to pH 10 (Figure 4) [60,61,62]. Additionally, pH gradients may be applied and adjusted during the experiment. SSME measurements using activating solutions containing different ion species enable the investigation of ion specificity. There is no need to change the overall ionic composition of the measurement buffers, which may be required in APC to reduce off-target conductivities, or loss of the giga-seal.

The high assay flexibility in SSME is also due to multiple options when selecting the target membrane. APC is currently limited to whole cells, while SSME can measure membranes from any sources, i.e., animal or plant tissues, liposomes with purified protein, or organelles such as mitochondria, endoplasmic reticulum, or lysosomes [33]. This makes SSME particularly intriguing for the study of intracellularly localized target proteins that are challenging to access by patch-clamp techniques.

In addition, the higher currents observed in SSME recordings allow for the measurement of slow transporters that are not readily accessible via APC. The signal-to-noise ratio is constrained by the number of target proteins that can be simultaneously measured. This is greatly influenced by the overexpression of the target protein in both APC and SSME techniques. Beyond the degree of overexpression of the target protein, the limiting parameter in APC is the surface area of the plasma membrane of the live cell. In SSME, the circular sensor surface of 3 mm in diameter represents a surface that is about 1000-fold larger. Consequently, the current amplitudes in SSME recordings are three orders of magnitude larger when the same driving forces are applied. We found average, control corrected currents from APC recordings (at 0 mV and ΔCs^+^ > 120 mM) and SSME recordings on the SURFE^2^R N1 (at 0 mV and ΔK^+^ = 50 mM, Figure 2D) of (0.104 ± 0.117) nA and (19.28 ± 1.22) nA, respectively. The current obtained with SSME is ~200-fold larger, despite a >2.5-fold lower driving force in our SSME assay and a sample amount that did not saturate the SURFE^2^R sensor surface (1:10 dilution of sample stock).

#### 3.6.6. APC Excels in Throughput Compared to SSME and LPC

Currently, APC holds the advantage in terms of throughput, as it enables the parallel measurement of up to 384 cells, whereas SSME is presently limited to the parallel recording of 96 sensors. Both APC and SSME environments offer automated preparations, measurement workflows, and data analysis capabilities. Developing a 384-well-based SSME instrument faces challenges due to the difficulty in accommodating ground electrodes, liquid handling components, and a 3 mm sensor surface to maintain a high signal-to-noise ratio into a smaller well. However, considering the relatively early stage of the technology, there is still potential for future advancements.

The difference in throughput between APC and SSME recordings decreases when the time to prepare the experiment is considered. In APC, a running cell culture is required, while in SSME samples may be stored at −80 °C for several months and sensors may be prepared in batches, both reducing the preparation time compared to APC. Although efforts to establish lysosomal APC is underway, it will still require fresh lysosomal preparations each day, essentially dropping its throughput compared to SSME.

Besides the parallelization of the measurements and the preparation time for sample and sensors, the number of sequential experiments on the same sample may also affect the throughput. Here, SSME shows the highest stability, essentially allowing for continuous measurements on the same sensor as long as the target protein is stable at room temperature [33]; this may enable stable measurements on the same sensor for 24 h or longer, as shown for EAAT3 recorded with the SURFE^2^R 96SE (Appendix A). Whole-cell APC requires stable giga-seals and stable access resistance to support long recordings, which can last for hours. To achieve such high stabilities, additions like fluoride or BSA may be required. Due to the generally lower stability of the giga-seals in LPC compared to whole-cell APC, the presented LPC datasets contained only four datapoints per sample: three compound concentrations and one control (Figure 8).

#### 3.6.7. APC and SSME Recordings Achieve Similar Data Quality

There are multiple routes to define and assess data quality, but when comparing the different aspects of the recorded data across APC and SSME experiments presented in this study, the overall data quality seems similarly high. SSME outperforms APC in terms of success rates, z’ prime values, and the quality of fits. However, APC is less susceptible to off-target compound effects.

(1) SSME recordings exhibit higher success rates (96.4 ± 3.3%, Figure 6C) compared to APC recordings (82 ± 5.2%, Figure 7D). This is attributed to the exceptional stability of SURFE^2^R sensors and the in vitro nature of the technique, whereas in APC recordings, the success rates are affected by variations between live cells and the stability of the giga-seal.

(2) The analysis of compound effects using both SSME (0.87 ± 0.027, Figure 6C) and APC (0.768 ± 0.058, Figure 7D) demonstrates high reliability, as indicated by the high z’ prime values. For calculating z’ prime, the standard error of the mean was employed to consider the number of data points [44]. Despite having fewer data points, SSME recordings exhibit slightly higher z’ prime values.

(3) The relative standard deviations of TMEM175 currents when averaging across cells in APC (±76%) and sensors in SSME (±24%) are somewhat lower in SSME recordings, due to the higher variability between live cells compared to the sensor-to-sensor variations. However, both techniques usually employ in-well normalization before averaging across multiple cells or sensors.

(4) The accuracy of EC_50_ fits is higher for SSME data (R^2^ = 0.978 ± 0.017) compared to APC data (R^2^ = 0.937 ± 0.054), indicated by overall higher values for the adjusted R² (Table 4), which is likely a consequence of lower variability of the data (see above). Consequently, relative errors of EC_50_ and I_max_ values determined from SSME data may be lower; however, direct comparisons are difficult since the different apparent potencies in APC and SSME will also affect the error of the determined parameters.

(5) Controls that endogenously express TMEM175 reveal artifact currents in SSME (−1 nA, Figure 1K) and leak currents in APC recordings (+1 nA, Figure 7), which are of similar relative magnitude compared to the TMEM175 sample currents recorded using both assays (+4 nA, Figure 1K and Figure 7). Increasing artifacts and leaks reduce the TMEM175-specificity of the read-out. In SSME experiments, artifact currents result from ion–lipid interactions and can be mitigated by adjusting the experimental conditions, as demonstrated in the case of TMEM175 by reducing the K^+^ concentration (Figure 3). However, the leak currents observed with APC result from different factors, including the size of the cells, giga-seal quality, and endogenous outward currents originating from a series of ion channels including the K^+^ channels, TRP channels, and purinergic channels commonly present in HEK cells, which may be addressed by adding channel blockers.

(6) Finally, compared to APC (Figure 7F), off-target compound effects seem to be more relevant in SSME recordings for some types of compounds starting at ~20 µM (Figure 8 and Figure 9F). The relevance of these effects for compound screening may be questioned, since poor compound affinities of 20 µM or above are likely not relevant for human treatment. Additionally, if necessary, such effects may be corrected through measurements on control membranes without the target protein.

### 3.7. Final Remarks

In this study, we conducted assays using SSME to demonstrate its ability to measure TMEM175. These assays successfully reproduced the ion selectivity and permeability ratios previously reported (Figure 2, Figure 3, Figure 4 and Figure 5). Additionally, leveraging the broad window of assay conditions provided by SSME, we made novel discoveries, including a comprehensive quantitative analysis of the impact of cytosolic and lysosomal pH levels on K^+^ flux through TMEM175 (Figure 4).

We also recorded multiple datasets using tool compounds (Figure 6 and Figure 7) and a set of blinded test compounds (Figure 8). Comparing the drug potency found with SSME, whole-cell APC, and LPC, we found that apparent drug potencies matched across technologies and assay conditions for some compounds, but showed major differences for others. The different results across methodologies may reveal important parameters that could affect the potency of a drug, such as the lipid and protein environment within the target membrane, the orientation of the protein, the membrane voltage, ion and pH gradients, and ion specificity. Most important are the differences between K^+^ and H^+^ fluxes.

To achieve substantial results, it is generally advantageous to employ an assay that closely mimics the native conditions. In this regard, LPC and SSME techniques have several advantages over whole-cell APC. On the other hand, we could not yet obtain pure lysosomes for SSME recordings and the physiological system in LPC is disrupted due to factors such as vacuolin-1 treatment. A major advantage of whole-cell APC stems from the notably higher throughput with up to 384 parallel recordings, surpassing that of LPC and maintaining a slight edge over SSME. Additionally, patch-clamp is the gold standard for drug development on ion channels and widely accepted by regulatory authorities such as the US Food and Drug Administration (FDA) and the European Medicines Agency (EMA), while SSME is a relatively novel technology. However, to date, around 150 peer-reviewed publications incorporating SSME recordings have been published, with several of them investigating drug transport [50,63] or how compounds modulate the activities of channels, transporters, and ion pumps [36,64,65]. Sanofi recently published detailed protocols on the utilization of SSME in early drug discovery [35]. To conclude, APC, SSME, and LPC serve as complementary technologies due to their distinct capabilities and advantages, with SSME filling a gap, enabling electrophysiological investigation with decent throughput on non-treated lysosomes without the requirement of a running cell culture.

## 4. Materials and Methods

### 4.1. Cell Culture

A human embryonic kidney (HEK293) cell line expressing TMEM175 under control of a tetracycline inducible expression system was used for the purpose of this project. As controls in the respective experiments, HEK293 UT cell lines were used for APC recordings, and a parental HEK293 cell line was employed for SSME recordings. The cells were maintained in Dulbecco’s Modified Eagle’s Medium (DMEM) media with 2 mM L-Glutamine, 10% Tetracycline Systems Approved fetal bovine serum (FBS), 5 µg/mL Blasticidin, and 2 mg/mL Geneticin. All cells were cultured in a humidified 5% CO_2_ incubator at 37 °C and maintained at 70–80% confluence while in culture. Prior to testing, cells were plated for 24, 48, or 72 h total and induced with tetracycline (1–3 µg/mL) for 24 h at 37 °C.

#### Western Blot

HEK293T TMEM175 stable cells were induced overnight and lysed in lysis buffer (5 mL Triton X-100 and 50 µL 100× protease/phosphatase inhibitor). The Pierce assay was used to determine protein concentration to ensure equal loading of 20 µg per well. Samples were denatured in 1x reducing agent (NuPAGE sample reducing agent: Cat. No: NP0009) and 4× Licor loading buffer (Cat. No: 102673–500) by heating samples to 95 °C for 5 min. The samples were then loaded into 4–12% bis-Tris gel (1.5 mm) and run in 1× MES running buffer for 90 min at 150 V. The gel was transferred with iBlot as per the manufacturer’s protocol. Antibody staining was performed by first blocking the gel using Licor block buffer at room temperature for 1 h followed by the incubation with the following primary polyclonal antibodies at 4 °C for 24 h: rabbit anti-TMEM175 (Proteintech 19925-1-AP, dilution 1:1000) and mouse anti-GAPDH (Abcam ab8245, dilution: 1:5000). Subsequently, the secondary polyclonal antibody Goat anti-rabbit (Li-Cor 926-32211, dilution: 1:5000) was incubated for 1 h at 4 °C.

### 4.2. SSME on Lysosomes

#### 4.2.1. Cell Harvest and Cell Disruption

For each preparation, cells from at least ten T175 flasks were harvested. Flasks were rinsed twice with harvesting buffer (10% PBS, 2 mM EDTA, 1 cOmplete^TM^ protease inhibitor cocktail per 50 mL), and then incubated for 10 min in harvesting solution at room temperature. Flasks were then penned until the cells detached. Cells were collected in a 15 mL tube, stored at 4 °C, and subsequently centrifuged at 26,000× *g* for 2 min. Each pellet was resolved in harvesting solution and centrifuged again at 26,000× *g* for 5 min. Pellets were frozen in liquid nitrogen before storage at −80 °C.

Cell disruption was performed with pre-cooled equipment and on ice. Cells were resuspended in 10 mL of disruption buffer (10 mM Tris, pH 7.4, 250 mM sucrose, 1 cOmplete^TM^ protease inhibitor cocktail per 50 mL) per gram of dry cell pellet. Cells were then transferred to the cell disruption vessel (Parr Instrument Company, Moline, IL, USA, Cat. No. 4639) and the volume was adjusted to a total of 30 mL. The cell disruption vessel was first incubated at 4 °C and 70 bar for 20 min, and then at room temperature for 5 min. The cell lysate was then released into a 50 mL Falcon tube by slowly opening the outlet valve of the cell disruption vessel. The cell lysate was directly processed to isolate lysosomes.

#### 4.2.2. Isolation of Lysosomes for SSME Recordings

The following procedure is based on the protocol developed by Schulz et al. [38], which was used to generate the datasets presented in this manuscript. The cell lysate was centrifuged for 10 min at 6000× *g* and 4 °C to remove cell debris (Beckmann Coulter, Brea, CA, USA, 70.1 Ti rotor). The supernatant was carefully transferred into an ultracentrifugation tube, followed by centrifugation for 30 min at 100,000× *g* and 4 °C to collect the membranes (Beckmann Coulter, 70.1Ti rotor). Membrane pellets were resuspended in up to 3 mL disruption buffer.

To enrich lysosomes, ultracentrifugation using a sucrose gradient was performed for 18 h at 100,000× *g* and 4 °C, using a swing-out rotor (Beckman Coulter, SW32 rotor). Then, 9%, 31%, 45%, and 70% (*w*/*v*) sucrose buffers were prepared in 10 mM Tris (pH 7.5). The resuspended membrane pellet was diluted in 70% sucrose buffer to achieve a final sucrose concentration of 51%. Ultracentrifugation tubes were stacked with the following solutions from bottom to top: resuspended membrane pellet (final sucrose 51%), 9 mL 45% sucrose, 9 mL 31% sucrose, 6 mL 9% sucrose.

After centrifugation, the tubes were placed on ice and three bands were collected in the following order: 9%/31% (upper band, plasma membrane fraction), 31%/45% (lower band, lysosomal fraction), 45%/51% (pellet band). The collected bands were 4x diluted in storing solution (30 mM HEPES, pH 7.5 (HCl), 140 mM NMDG-Cl, 2 mM MgCl_2_, 0.2 mM DTT, 5% glycerol) and subsequently centrifuged for 30 min at 100,000× *g* and 4 °C (Beckmann Coulter, 70.1Ti rotor). Pellets were resuspended in ~200 µL storing solution, and 10 µL aliquots were frozen in liquid nitrogen before storage at −80 °C.

For sample validation (Figure 1B), we also applied a procedure based on the protocol developed by Jinn et al. [5]. The cell lysate was centrifuged for 5 min at 500× *g* and 4 °C to remove cell debris. The supernatant was then centrifuged for 10 min at 6800× *g* and 4 °C to remove mitochondria, and then centrifuged again for 30 min at 20,000× *g* and 4 °C to collect the lysosomes. The pellet was gently resolved in 20 mL sucrose buffer (0.25 M sucrose, 20 mM HEPES, pH 7.4, 1 cOmplete^TM^ protease inhibitor cocktail per 50 mL) and centrifuged for 15 min at 20,000× *g* and 4 °C. The pellet was resolved in ~200 µL storing buffer, aliquoted, and frozen as explained above.

One aliquot of each sample was used to determine the total protein concentration using a Bradford assay, which typically was between 1 mg/mL and 10 mg/mL, depending on the sample batch.

#### 4.2.3. ELISA

To determine the efficacy of fractionation and membrane enrichment procedures, the lysosomal marker LAMP1 and the plasma membrane marker Na-K-ATPase were quantified using quantitative sandwich ELISA kits (Human Sodium/Potassium-Transporting ATPase Subunit Alpha-1(ATP1A1) ELISA Kit, Cusabio, Houston, TX, USA, CSB-EL002322HU; Human LAMP1 ELISA Kit, Biorbyt, Cambridge, UK, Orb565022) according to the manufacturer’s instructions.

Briefly, microplate wells were already coated with either anti-LAMP1 or anti-ATP1A1 antibodies. Then, 10 µL of each sample was diluted with 90 µL of assay diluent. Standards and samples were pipetted into the wells in duplicates and any target protein became bound by the immobilized antibody. After removing any unbound substances, a biotin-conjugated antibody specific for the target protein was added to the wells. After washing, avidin-conjugated horseradish peroxidase (HRP) was added to the wells. Following a wash to remove any unbound avidin enzyme reagent, a substrate solution was added to the wells and color developed in proportion to the amount of target protein bound in the initial step. The color development was stopped, and the optical density of each well at 450 nm was measured using a microplate reader. A standard curve was generated by five parameter logistic (5-PL) curve-fit and the sample concentration was calculated by interpolation of the sample OD into the standard curve.

#### 4.2.4. SSME Recordings

Automated SSME recordings were performed on the SURFE^2^R N1 and the SURFE^2^R 96SE (Nanion Technologies), using the SURFE^2^R N1 Control 1.7.0.2 and the SURFControl96 1.7 software, respectively.

Technical details of both devices have been reviewed recently [33,34] and are summarized in the Appendix A. In brief, both instruments perform a fast solution exchange from non-activating solution to activating solution at 0 mV, applying a substrate gradient to activate charge translocation of TMEM175 in lysosomes, which are immobilized on a gold-coated sensor chip. The SURFE^2^R N1 is a single-well device, while the SURFE^2^R 96SE enables high-throughput sequencing by utilizing 96-well sensor plates, facilitating 96 simultaneous recordings. Although the devices employ different liquid handling methods and electrophysiological hardware, the measurement principle, experimental conditions, and workflows used are fundamentally identical.

#### 4.2.5. Sensor Preparation

For SSME recordings on the SURFE^2^R N1, we used 3 mm sensors; the SURFE^2^R 96SE employs 96-well sensor plates. A total of 50 µL of 0.1 mM octadecanethiol in isopropanol was added to each sensor well, following incubation for 30 min at room temperature to form an alkanethiol monolayer on the sensor surface. Subsequently, the sensors were washed twice with isopropanol and three times with ddH_2_O, before dry-tapping on a tissue. The subsequent process of sensor coating with lipid and lysosomes was carried out manually for the SURFE^2^R N1 and automated with the SURFE^2^R 96SE serving as a pipetting robot. In both cases, 1.5 µL of 7.5 mg/mL 1,2-diphytanoyl-sn-glycero-3-phosphocholine (DPhPC) in n-decane was added to the center of the circular gold surface, followed by the addition of 50 µL of non-activating solution to form the SSM. The composition of the non-activating solution was adjusted for each experiment (Table 2). It consisted of 30 mM HEPES, 30 mM MES, 5 mM MgCl_2_, and 140 mM NaCl at pH 7.6 for our standard K^+^ conductivity assay. SSM-coated 96-well sensor plates were stored at −80 °C before use.

Lysosomes were thawed and diluted in non-activating solution to achieve a total protein concentration between 40 µg/mL and 0.8 mg/mL—depending on the signal-to-noise requirements of the respective experiment—and sonicated using a tip sonicator (UP 50 H, Dr. Hielscher, Teltow, Germany, equipped with MS 1 tip; 10 bursts, 20% amplitude, 0.5 s cycle time). Then, 9 µL (SURFE^2^R 96SE) or 10 µL (SURFE^2^R N1) of the processed lysosomes were added to each sensor well by pipetting to the non-activating solution inside the sensor wells, while submerging the pipette into the solution, close to the sensor surface. The sensors were then centrifuged for 30 min at 2500× *g* at room temperature to yield stable adsorption of lysosomes onto the sensors. To start a sequence of solution exchange measurements, single sensors and 96-well sensor plates were placed into the measurement chamber of the SURFE^2^R N1 or SURFE^2^R 96SE instrument, respectively.

#### 4.2.6. Measurement Solutions

In a typical single-solution exchange SSME experiment, the non-activating solution (NA) is exchanged with the activating solution (A), which contains the ion that is translocated through TMEM175. The NA-A exchange simultaneously provides the driving force and stimulates the charge translocation through TMEM175.

When the impact of an additional ion gradient is investigated, i.e., when investigating the effect of a pH gradient on the K^+^ conductivity (Figure 4G), the resting solution (R) defines the composition of the lysosomal lumen. The R-NA solution exchange establishes the ion gradient, followed by the NA-A solution exchange that provides the driving force which stimulates charge translocation through TMEM175 in the presence of the additional ion gradient.

Depending on the assay, different measurement solutions are used. NA, A, and R used for the same experiment are prepared from the same stock of main solution (M). The compositions for all solutions are provided in Table 2. If not stated otherwise, we applied a solution exchange from 50 mM Na^+^ (NA) to 50 mM K^+^ (A) to measure the baseline current of TMEM175 activity. For compound assays, a serial dilution of the compound was prepared in DMSO, then each compound stock was added to NA and A at a dilution factor of 1:1000, achieving a final 0.1% DMSO concentration in all solutions.

#### 4.2.7. Electrophysiological Recordings and Data Analysis

The solutions were injected to the sensor wells with a flow speed of 200 µL/s for both recordings with the SURFE^2^R N1 and the SURFE^2^R 96SE. The SURFE^2^R N1 enables a continuous solution flow across the sensor. We used a flow protocol comprising 1 s NA, 1 s A, and 1 s NA solution, followed by rinsing the sensor with 1 mL NA or R, depending on the assay. For the SURFE^2^R 96SE, the injection and removal of measurement solutions occurred sequentially. First, a stack of 80 µL NA, 50 µL A, and 60 µL NA was added to the sensor. Then the solution was removed, followed by rinsing of the sensor with 200 µL NA or R, depending on the assay.

During the NA-A-NA exchange, the current was recorded using a sampling rate of 1 kHz. If not stated otherwise, the peak current observed after NA-A exchange was used for subsequent analysis. The following QC conditions were applied to each sensor within the SURFE^2^R 96SE plate: minimum sensor capacitance of 4 nF; minimum sensor resistance of 100 MΩ; minimum reference signal amplitude of 1 nA; stable current during the reference measurement, typically a sequence of three activations using 50 mM K^+^.

Datasets recorded with the SURFE^2^R N1 are usually obtained with the same sensor, i.e., all concentration-dependent datapoints are obtained by sequential measurements. Before averaging the datasets across sensors, the datasets were normalized. If not stated otherwise, the datasets recorded with the SURFE^2^R 96SE were obtained from different sensors. Each sensor was used to record the TMEM175 baseline current (Figure 6B). Subsequently, a measurement using one assay condition (compound/ion concentration, pH) was performed. Before averaging across sensors, the obtained datapoint was normalized to the baseline current recorded with the same sensor. The process of data analysis, including artifact correction and normalization for the different datasets presented in the results section, is outlined in Appendix A.

### 4.3. Whole-Cell APC

#### 4.3.1. Cell Harvest

Cell confluency was 60–80% at the time of harvest. The cells were detached from the flask using TrypLEExpress at 37 °C. Subsequently, cold divalent-free extracellular solution was added to the flask, followed by incubation at 4 °C and, finally, trituration of the cells. The cell solution was then placed on a cell hotel at 10 °C and a shake speed of minimum 200 rpm, located on the SyncroPatch platform, for at least 30 min prior to testing.

#### 4.3.2. Whole-Cell APC Recordings

Automated patch-clamp recordings were performed using the SyncroPatch 384i (Nanion Technologies). The voltage protocol generation and data collection were performed with PatchControl384 v1.9.0 and analyzed with DataControl384 v2.2.0. A multi-hole, medium resistance chip with four patch-clamp holes per well was used for recording.

The following QC conditions were applied to each well: minimum seal resistance of 10 MΩ (for tool compounds, Figure 7) or 50 MΩ (for test compounds, Figure 8); maximum series resistance of 20 MΩ; minimum current amplitude during control period of 200 pA; stable current during the external solution application (2nd liquid period); inhibition with reference inhibitor 4-AP to verify the presence of functional TMEM175.

#### 4.3.3. Cs^+^ Flux Assay

The measurement solutions for the Cs^+^ flux assay as used for compound measurements had the following compositions. The external solution representing the lysosomal lumen contained 140 mM NaCl, 4 mM KCl, 2 mM CaCl_2_, 1 mM MgCl_2_, 10 mM HEPES, 5 mM D-glucose, and pH 7.4 (NaOH). DMSO remained constant at 0.2% and BSA at 0.05% in the external solution throughout the assay, if not stated otherwise. The internal solution representing the cytosol contained 110 mM CsF, 10 mM CsCl, 10 mM NaCl, 10 mM EGTA, 10 mM HEPES, 2 mM NaATP, and pH 7.2 (CsOH).

The voltage protocol applied for the investigation of the tool compounds (Figure 7) consisted of a 500 ms ramp from −80 mV to +80 mV, followed by a 400 ms step from −80 mV to +80 mV before returning to the holding potential of 0 mV (Figure 7B). The maximum outward current amplitude obtained at +80 mV from the ramp section was used for analysis. To investigate the blinded test compounds (Figure 8), an alternative voltage protocol was applied. It consisted of a 125 ms ramp from −120 mV to +100 mV from a holding potential of −90mV, followed by 40 mV and 60 mV voltage steps from +100 mV to −90 mV. The maximum outward current amplitude obtained at 0 mV from the voltage step section was used for analysis. In both voltage protocols, voltage sweeps were repeated every 10 s and currents sampled at 10 kHz. We applied both voltage protocols to a subset of compounds and found that the results were identical.

The application protocol for agonists consisted of two additions of extracellular physiological solution for a minimum of 3 min, followed by the external application of the test compound for a minimum of 2 min (Figure 7C). Afterwards, 0.5 mM (Figure 7) or 0.3 mM (Figure 8) ZnCl_2_ was applied as a reference activator, before the channel was inhibited by a saturating concentration of 4-AP (1 mM). The application protocol for antagonists consisted of two additions of extracellular physiological solution for a minimum of 3 min, followed by an application of the test compound for a minimum of 2 min, and then the channel was inhibited by a saturating concentration of 4-AP (1 mM).

Currents recorded after compound addition were normalized to the baseline current before compound addition, averaged, and plotted over the compound concentration to determine the EC_50_ and IC_50_ values.

#### 4.3.4. H^+^ Flux Assay

The measurement solutions for the H^+^ conductivity mode had the following compositions. The external solution representing the lysosomal lumen contained 140 mM NMDG, 80 mM MSA, 7.5 mM CaCl_2_, 1 mM MgCl_2_, 20 mM MES and 0.675 mM HEPES, pH 6.0 (MSA). DMSO remained constant at 0.2% and BSA at 0.05% in the external solution throughout the assay. The internal solution representing the cytosol contained 20 mM CsF, 90 mM NMDG, 10 mM CsCl, 10 mM NaCl, 10 mM EGTA, 10 mM HEPES, and pH 7.2 (CsOH). Compared to the Cs^+^ flux assay (Section 4.3.3), cations in the external solution that have been described to permeate through TMEM175 were replaced by NMDG, and the external pH was acidified, mimicking the lysosomal environment of TMEM175.

We used a similar voltage protocol as for the Cs^+^ flux assay, applying a 500 ms voltage ramp from −100 mV to +100 mV, followed by a voltage step to −100 mV, which was held for 1 s, before returning to the holding potential of 0 mV. Each voltage sweep was repeated every 10 s and currents sampled at 10 kHz. The maximum inward current amplitude obtained at −100 mV from the ramp section was used for analysis.

The application protocol for the pH assay consisted of two to three additions of extracellular physiological solution at pH 7.2 for a minimum of 3 min, followed by an application of activating pH condition (pH 6.0), or control condition (pH 7.2), for a minimum of 3 min. The test compound was then applied before the channel was inhibited by a saturating concentration of 4-AP (1 mM).

### 4.4. Lysosomal Patch-Clamp

#### 4.4.1. Preparation of Lysosomes

Experiments were carried out in whole-lysosome configuration in voltage-clamp mode, using a modified patch-clamp method, as described by Chen et al. [30].

Briefly, cells were treated overnight with 5 µM vacuolin-1 to increase the size of the lysosomes. Mechanical isolation was performed using a patch pipette pressed against a selected cell, and quickly pulled away to slice the cell membrane to release the enlarged lysosomes into the dish. After the dissection of the lysosomes, the glass electrode was exchanged for a recording electrode. After the formation of a giga-seal and compensation of capacitance transients, voltage steps of several hundred millivolts with millisecond duration were applied to break into the vacuolar membrane.

#### 4.4.2. Lysosomal Patch-Clamp Recordings

The bath solution representing the cytosol contained 10 mM HEPES (pH 7.2), 145 mM K-methanesulfonate and 5 mM KCl. The pipette solution representing the lysosomal lumen contained 10 mM MES (pH 5.5), 145 mM Cs-methanesulfonate, 5 mM KCl, and 10 mM EGTA.

After the reappearance of capacitance transients to verify the whole-lysosome configuration, the TMEM175 currents were recorded using a ramp protocol from −100 to +100 mV with 1 s of duration, digitized at 20 kHz. The ramp protocol was repeated in 4 s intervals, keeping the lysosomes at 0 mV in between. We used the outward current at +100 mV as a read-out, essentially observing the flux of H^+^ and Cs^+^ from the lysosome into the cytosol.

The compound application protocol consisted of five applications, starting with bath solution (control), followed by bath solution in the presence of the compound. Three compound concentrations were applied in ascending order, and the subsequent concentrations were added when the steady state was reached. The compounds were perfused continuously to obtain an I/t-plot of the current during the compound application. When the seal stability was sufficient, we added 100 µM AP-6 after the compound addition sequence to obtain a full block of the current.

Before averaging across datasets obtained from N = 3 lysosomes, we normalized the currents to the control current. Datasets containing 6 or 9 compound concentrations were recorded in two or three sets of experiments with 3 compound concentrations per lysosome. All experiments were conducted at room temperature, and recordings were analyzed with pClamp 10.7 and GraphPad Prism 8.

## Figures and Tables

**Figure 8 ijms-24-12788-f008:**
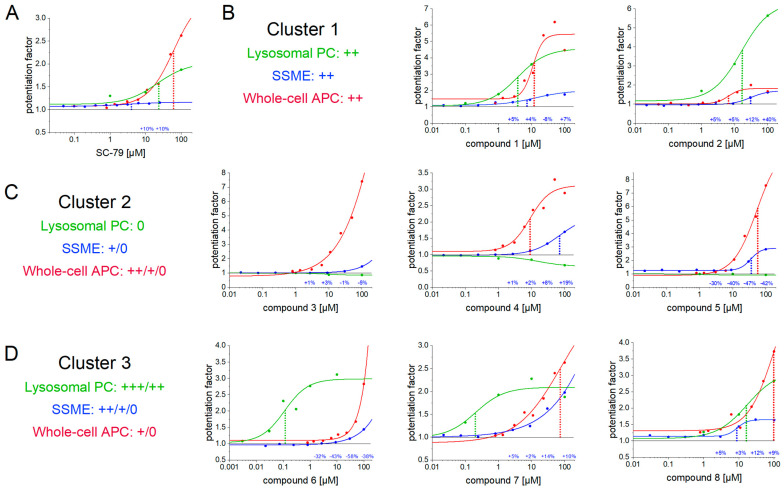
Effects of eight blinded test compounds and SC-79 on TMEM175 recorded with whole-cell APC, manual LPC, and SSME. Compounds are grouped into three clusters based on apparent EC_50_ values, as indicated: EC_50_ < 1 µM (+++), EC_50_ = 1–30 µM (++), EC_50_ = 30–100 µM (+), EC_50_ > 100 µM (0); (**A**) Dose-response curves for SC-79; (**B**) Dose-response curves for cluster 1 compounds showing similar effects with all three technologies; (**C**) Dose-response curves for cluster 2 compounds showing no activity in LPC, but higher activity in APC; (**D**) Dose-response curves for cluster 3 compounds showing high activity in LPC, but lower activity in APC; For all assay technologies, datapoints were normalized to the current recorded before compound addition and averaged across at least N = 3 individual experiments. For clarity reasons, the SEM is not shown; the average relative SEM after averaging datapoints recorded with two highest compound concentrations is 35.6% for LPC data and 2.9% for SSME data. The APC data were acquired using a voltage application protocol distinct from the one depicted in Figure 7B, as described in the Methods section. Datasets were not corrected for artifacts (as observed in SSME) or off-target compound effects. For SSME data, we indicate the difference in the peak current when applying the four highest concentrations of a compound to control lysosomes (to assess off-target effects) compared to the current recorded with TMEM175 lysosomes prior to compound addition. Positive values represent off-target potentiation (potentially overestimating I_max_), negative values represent off-target inhibition. Considerable off-target compound effects are observed when the given percentage approaches the percentage of enhancement recorded with the TMEM175 sample at the given compound concentration. All datasets were fitted using the EC_50_ equation I = I_max_ − (I_max_ − I_min_)/(1 + (c/EC_50_)^n^). When an EC_50_ value was determined from the fit, it is highlighted by a vertical dotted line. For LPC data, the Hill coefficient n was fixed to 1 due to the lower number of datapoints per dataset, which is typically three. The obtained fit parameters are shown in Table 4.

**Table 1 ijms-24-12788-t001:** Comparison of assay and hardware related parameters in SSME recordings on TMEM175 using the SURFE^2^R N1 and the SURFE^2^R 96SE.

Parameter	SURFE^2^R N1	SURFE^2^R 96SE
Assay parameter		
Main buffer (M) ^1^	30 mM HEPES, 30 mM MES, 5 mM MgCl_2_, 90 mM NaCl, pH 7.6 (NaOH)	
Non-activating solution (NA) ^1^	M + 50 mM NaCl	
Activating solution (A) ^1^	M + 50 mM KCl	
Total protein concentration of the sample	2.2 mg/mL	
Sample dilution used per sensor well	1:10	1:100
Sample volume per sensor	10 µL	8 µL
Sample consumption based on total protein concentration determined via Bradford assay	2.2 µg protein per sensor	0.18 µg protein per sensor; 17 µg protein per plate
Typical mode of measurement	Sequential recordings on the same sensor using different buffer compositions (i.e., pH or concentration sequence); first, three measurements using standard conditions; in-well normalization of dataset to standard activation response before averaging across sensors	Parallel recordings on different sensors; first, three measurements using standard conditions; then, one assay condition is applied per sensor (i.e., pH, ion concentration, compound addition); in-well normalization of single data point to standard activation response before averaging across sensors
Hardware parameter		
Parallelization	Single well	96 wells
Sensor diameter	3 mm	3 mm
Liquid handling of solution exchange	Continuous solution flow: 1 s NA, 1 s A, 1 s NA	Stack of solutions in 200 µL pipette: 50 µL NA ^2^, 30 µL A, 80 µL NA
Restoring initial conditions	Rinse with 1 mL NA at continuous flow conditions	Stepwise dilution: remove solution down to ~30 µL well volume; rinse with 200 µL NA; remove solution down to ~30 µL well volume; refill with 60 µL NA
Solution flow speed (defines time resolution)	200 µL/s	200 µL/s
Time resolution taken from [33]	38 ms	7 ms
Measurement time window (time of A flow)	1 s	~250 ms
A consumption per measurement	300 µL (200 µL + 100 µL spare)	50 µL
NA consumption per measurement	1.6 mL (Measurement: 600 µL; rinse: 1 mL)	~0.4 mL (Measurement: 130 µL; rinse + refill: 260 µL)
Solution consumption per compound and well in compound assay	~2 mL (rinse, incubate, measure)	~0.4 mL (rinse, incubate, measure)
Read-out characteristics		
TMEM175 peak current	(12 ± 3) nA	(4.1 ± 1) nA
Control peak current	(−1.07 ± 0.3) nA	(−1.29 ± 0.52) nA
TMEM175 rise time constant τ_1_	(5.1 ± 1.3) ms	(9.2 ± 1.7) ms
TMEM175 decay time constant τ_2_	(10.7 ± 1.1) ms	(10.2 ± 0.8) ms
TMEM175 system time constant τ_3_ ^3^	(285 ± 71) ms	N.A.

^1^ This buffer composition represents our standard condition, which is used to examine the modulation of K^+^ flux through TMEM175, i.e., in our compound assays. ^2^ The first NA represents the initial solution, which is taken from the sensor well instead of the reservoir. This compensates for slight differences of the well solution compared to NA solution in the reservoir and prevents solution exchange artifacts. ^3^ The system time constant is a consequence of capacitor discharging and affected by capacitance and conductance of the compound membrane on the sensor [32]. It is only visible for fast currents in SSME and does not provide information about TMEM175, but rather represents properties of the sensor.

**Table 2 ijms-24-12788-t002:** Measurement solutions employed in SSME assays.

Assay	Main Solution (M)	Non-Activating Solution (NA)	Activating Solution (A)	Resting Solution (R)
Standard Assay (Figure 1G,K)	30 mM HEPES, 30 mM MES, 5 mM MgCl_2_, 90 mM NaCl, pH 7.6 (NaOH)	M + 50 mM NaCl	M + 50 mM KCl	--
Ion selectivity (Figure 2)	30 mM HEPES, 30 mM MES, 5 mM MgCl_2_, 90 mM NMDG-Cl, pH 7.6 (NMDG)	M + 50 mM NMDG-Cl	M + 50 mM Cl- salt (Li, Na, K, Rb, Cs, choline)	--
K^+^ translocation (I/c curve, Figure 3)	30 mM HEPES, 30 mM MES, 5 mM MgCl_2_, 90 mM NaCl, pH 7.6 (NaOH)	M + 300 mM NaCl	M + x mM KCl + (300 − x) mM NaCl (1 ≤ x ≤ 300 mM)	--
pH dependence, K^+^ flux (no ΔpH, Figure 4A–D)	30 mM HEPES, 30 mM MES, 5 mM MgCl_2_, 90 mM NaCl	M + 50 mM NaCl titrated to pH 3.0–10.0 (NaOH, HCl)	M + 50 mM KCl titrated to pH 3.0–10.0 (NaOH, HCl)	--
pH dependence, K^+^ flux (with ΔpH, Figure 4E–G)	30 mM HEPES, 30 mM MES, 5 mM MgCl_2_, 90 mM NaCl	M + 50 mM NaCl titrated to pH 7.5 (NaOH)	M + 50 mM KCl titrated to pH 7.5 (NaOH)	M + 50 mM NaCl titrated to pH 3.0–10.0 (NaOH, HCl)
H^+^ translocation (influx, Figure 5A–E)	30 mM HEPES, 30 mM MES, 5 mM MgCl_2_, 140 mM NaCl	M titrated to pH 7.6 (NaOH)	M titrated to pH 7.4–4.6 (NaOH)	--
H^+^ translocation (efflux, Figure 5F–J)	30 mM HEPES, 30 mM MES, 5 mM MgCl_2_, 140 mM NaCl	M titrated to pH 4.6 (NaOH)	M titrated to pH 4.8–7.6 (NaOH)	--
Compound IC_50_/EC_50_ (Figure 6)	30 mM HEPES, 30 mM MES, 5 mM MgCl_2_, 90 mM NaCl, pH 7.6 (NaOH)	M + 50 mM NaCl + x µM compound (0.1% DMSO)	M + 50 mM KCl + x µM compound (0.1% DMSO)	--

**Table 3 ijms-24-12788-t003:** Half saturation (EC_50_ or IC_50_) and current potentiation/inhibition of five tool compounds (4-AP, Zn, DCPIB, arachidonic acid) investigated with whole-cell APC and SSME. The indicated currents are relative currents in percent compared to the TMEM175 baseline current (100%). In the case of enhancers, the current represents the I_max_ of the dose–response curve; in the case of inhibitors, the current reflects the inhibition found with the highest compound concentration, which is indicated. All compound data recorded using SSME are shown in Figure 6; compound data recorded with APC are shown in Figure 7. In SSME data, the average artifact current recorded with control lysosomes without TMEM175 overexpression in the absence of compound was subtracted from all datasets before data averaging and fitting. Off-target compound effects were not considered.

		4-AP	Zn	DCPIB	Arachidonic Acid
Half saturation	SSME	(1.46 ± 0.3) mM	(1.53 ± 0.21) mM (inhibitor)	(9.6 ± 5.1) µM	(2.25 ± 0.45) µM
	APC	(16.6 ± 8.2) µM	(102 ± 12) µM (activator)	(16.7 ± 11.3) µM	(83 ± 20) µM
Relative current	SSME	1.0 ± 3.6% (10 mM)	19.2 ± 4.6% (10 mM)	275 ± 37% (60 µM)	168 ± 5% (50 µM)
	APC	−24 ± 2% (50 µM)	400 ± 40% (50 µM)	579 ± 105% (50 µM)	137 ± 6% (30 µM)

**Table 4 ijms-24-12788-t004:** Summary of data obtained for the eight blinded test compounds and SC-79 using whole-cell APC, manual LPC, and SSME, as shown in Figure 8. All parameters were derived from the respective fits using the EC_50_ equation I = I_max_ − (I_max_ − I_min_)/(1 + (c/EC_50_)^n^). For all fits, the adjusted R² is provided as a measure for the quality of the fit.

Cluster	Compound	Technique	EC_50_ (µM)	n	I_max_ − I_min_ (nA)	R²
N.A.	SC-79	APC	61.8 ± 43.2	1.17 ± 0.38	2.45 ± 1.02	0.983
		SSME ^1^	2.9 ± 2.3 ^1^	0.89 ± 0.6	0.09 ± 0.04 ^1^	0.893
		LPC	21 ± 32	1	0.91 ± 0.52	0.733
1	**1**	APC	10.8 ± 3.1	2.78 ± 2.1	3.96 ± 1.04	0.846
		SSME	12.9 ± 15.1	0.86 ± 0.34	0.91 ± 0.41	0.971
		LPC	3.8 ± 1.2	1	3.51 ± 0.42	0.993
	**2**	APC	6.5 ± 2.7	2.38 ± 1.45	0.81 ± 0.18	0.88
		SSME	30.3 ± 12.8	1.61 ± 0.81	0.74 ± 0.2	0.943
		LPC	15.9 ± 5.8	1	5.16 ± 0.7	0.981
2	**3**	APC	no saturation	N.A.	6.4 ^2^	0.984
		SSME	no saturation	N.A.	0.46 ^2^	0.969
		LPC	no effect	N.A.	N.A.	N.A.
	**4**	APC	9 ± 3.3	1.43 ± 0.85	2.02 ± 3.63	0.905
		SSME	70.4 ± 16.4	1.04 ± 0.08	1.21 ± 0.14	0.999
		LPC	no effect	N.A.	N.A.	N.A.
	**5**	APC	54 ± 28	1.21 ± 0.33	9.72 ± 3.09	0.986
		SSME	34.8 ± 4.2	2.7 ± 1.1	1.67 ± 0.17	0.993
		LPC	no effect	N.A.	N.A.	N.A.
3	**6**	APC	no saturation	N.A.	1.73 ^2^	0.984
		SSME	no saturation	N.A.	0.48 ^2^	0.984
		LPC	0.11 ± 0.07	1	1.95 ± 0.43	0.887
	**7**	APC	73 ± 212	0.66 ± 0.54	3.25 ± 3.67	0.963
		SSME	no saturation	N.A.	0.98 ^2^	0.976
		LPC	0.2 ± 0.2	1	1.1 ± 0.35	0.83
	**8**	APC	96 ± 181	1.19 ± 0.77	4.75 ± 5.66	0.946
		SSME	9 ± 0.8	3	0.51 ± 0.03	0.985
		LPC	17.4 ± 6.6	1	2.07 ± 0.29	0.98

^1^ Effect of SC-79 in SSME reflects off-target effects that are also present in the control (Figure 9F). In SSME, missing effects of SC-79 are explained due to the absence of PKB [48]. ^2^ Since no saturation was observed, I_max_ could not be determined. Instead of I_max_ − I_min_, we provide I(100 µM) − I_min_.

**Table 5 ijms-24-12788-t005:** Comparison between TMEM175 assays and general technical limitations in SSME, whole-cell APC, and manual LPC.

	SSME	Whole-Cell APC	Lysosomal Patch-Clamp
Experimental conditions			
Driving force	Stimulus: concentration jumps; no direct voltage control, but voltage may be applied via uncouplers [55]	Stimulus: voltage steps; additional ionic gradients	Stimulus: voltage steps; additional ionic gradients
Solutions	Wide range of assay conditions, including pH gradients, extreme pH values and non-native ionic concentrations	Limited by the requirements of live cells; potentially requires chemicals to increase giga-seal stability, i.e., Fluoride, BSA; or channel blockers to increase specificity of the currents for the target protein	Potentially requires chemicals to increase giga-seal stability, i.e., fluoride, BSA, or channel blockers to increase specificity of the currents for the target protein
Target membrane	Lysosomes, non-treated, stored at −80 °C; impurities with plasma membrane vesicles	Plasma membrane of live cells	Freshly isolated lysosomes, chemically pre-treated using vacuolin-1 for enlargement
Protein orientation	Right-site-out	Inside-out	Right-site-out
Technical limitations			
Read-out	Capacitive currents; peak represents initial translocation rate; potential pre-steady-state currents triggered by ion/substrate binding [40]	Steady-state currents; potential pre-steady-state currents triggered by voltage steps	Steady-state currents; potential pre-steady-state currents triggered by voltage steps
Signal-to-noise	Up to 1000-fold larger currents compared to whole-cell APC due to large circular sensor surface (Ø 3 mm)	Signal-to-noise is limited by surface of the cell and the expression of the target protein inside the target membrane	Signal-to-noise is limited by the surface of the lysosome and the expression of the target protein inside the target membrane
Signal interpretation	Signal represents flux of the ion species provided during fast solution exchange	Voltage steps stimulate the flux of all available ions across the membrane	Voltage steps stimulate the flux of all available ions across the membrane
Control experiments reveal…	Solution exchange artifacts due to membrane-ion interaction [41]; off-target compound effects using high compound concentrations	Off-target leak currents; off-target compound effects potentially smaller compared to SSME	Off-target leak currents; off-target compound effects
Ease of use	Easy, overall process is automated	Easy, overall process is automated	Highly skilled technician required
Throughput	96 parallel recordings; lysosomes stored for months at −80 °C; sensor preparation in batches; stable sequential recordings	384 parallel recordings; running cell culture required; duration of recordings limited by giga-seal stability	No parallel recordings; fresh lysosomal preparations each day; duration of recordings limited by giga-seal stability, further reduced compared to whole-cell APC
Data quality	Ultra-high success rates (96.4 ± 3.3%); superior z’ prime (0.87 ± 0.0278); low standard deviation between sensors	High success rates (82 ± 5.2%); superior z’ prime (0.768 ± 0.058); high standard deviations between cells	Low current amplitudes and lower success rates compared to APC and SSME; higher standard deviation between lysosomes
Novelty	Not yet standard in drug screening	Well-established standard method	Gold standard for lysosomal channels

## Data Availability

The data presented in this study are available in the article and Appendix A.

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
