# Peer review of "A Comparative Study on the Lysosomal Cation Channel TMEM175 Using Automated Whole-Cell Patch-Clamp, Lysosomal Patch-Clamp, and Solid Supported Membrane-Based Electrophysiology: Functional Characterization and High-Throughput Screening Assay Development"

_ijms, 2023, doi:10.3390/ijms241612788_

Round 1
Reviewer 1 Report
Manuscript entitled “A comparative study on the lysosomal cation channel TMEM175 using automated whole-cell patch-clamp, lysosomal patch-clamp and SSM-based electrophysiology: Functional characterization and HTS assay development.” describes the in comparison of TMEM175 current with using various patch clamp technique. The current data are meaningful and lots of information and authors kindly addressed detailed description. However, here I provide several comments as below. The manuscript preparation and results needs to be greatly improved before it can be considered for publication.
1. In introduction section, the authors too broadly state. Specify the main theme and short information. Explanation of various patch clamp technique will be discussed in discussion section.
2. How to determine the TMEM175 expression? Although author showed Fig. 1B, author should perform several experiments. E.g. Western blot, Co-staining with lysosomal marker such as LAMP-2, to visualize the target protein TMEM175.
3. If TMEM175 was overexpressed, protein expression data of TMEM175 should be needed.
4. Enlarged lysosome by vacuolin-1 treatment will be useful technique to measure the lysosomal current.
5. In several figures, such as Fig 1 or 8, it is hard to follow because of too small size and lots of experimental conditions. Make simplify.
6. In the discussion section (3.7 Final remarks), authors should mention the finding of current study in the beginning.
Reviewer 2 Report
In this work entitled “A comparative study on the lysosomal cation channel TMEM175 using automated whole-cell patch-clamp, lysosomal patch-clamp and SSM-based electrophysiology: Functional characterization and HTS assay development” of Bazzone et al., they compare results across differebt assay technologies (whole-cell automated patch-clamp (APC), solid supported membrane based electrophysiology (SSME) and lysosomal patch-clamp (LPC)) discussing their advantages and disadvantages during the study of lysosomal cation channel TMEM175 involved in Parkinson’s Disease.
First of all, I am pleased for having had the opportunity to read this paper which, as the authors write, has two main objectives: enhance the understanding of the physiological function of TMEM175 and to identify suitable assay conditions for conducting electrophysiological recordings on this kind of compounds. The second objective is certainly very important nowadays and I appreciate the effort of having created a work of which a good part is centered on finding possible solutions on this topic.
But from my point of view it is precisely this part that makes this work very challenging to read, making it difficult to follow for those who are not insiders.
I would suggest to the authors to improve the presentation of their work, making the results part as lean as possible by summarizing the more technical and methodology parts and/or moving them to the "Materials and methods" section which I would also make a supplementary section.
Round 2
Reviewer 1 Report
The authors have made the revisions requested by the previous comments and the necessary corrections. However, I cannot reach the supplementary file. Please re-upload the different type of file such as pdf file.